# Evaluating the performance of CE-QUAL-W2 version 4.5 sediment diagenesis model

Manuel Almeida[1], Pedro Coelho[1]

[1]MARE - Marine and Environmental Sciences Centre, ARNET - Aquatic Research Network Associate Laboratory, NOVA School of Science and Technology, NOVA University Lisbon, Caparica, 2896-516, Portugal

*Correspondence to*: Manuel Almeida (mcvta@fct.unl.pt)

**Abstract.** This research evaluates the performance of the CE-QUAL-W2 v4.5 sediment diagenesis model in simulating water temperature, dissolved oxygen, total phosphorus, total nitrogen, chlorophyll-a, and biochemical oxygen demand in a Portuguese reservoir over a six-year period (2016–2021). The model was calibrated using 35 observed profiles of temperature and dissolved oxygen, as well as six annual measurements of total nitrogen, total phosphorus, chlorophyll-a, and biochemical oxygen demand at multiple depths. To benchmark performance, three alternative sediment oxygen demand formulations—a Zero-order, First-order, and a Hybrid model combining both approaches—were also implemented and compared. All models achieved NSE and RMSE values within or near the ranges reported in the literature, effectively capturing the system's water quality dynamics. Among them, the Hybrid model yielded the best overall performance while maintaining a simpler structure (Water temperature - NSE: 0.96±0.18; RMSE: 1.09±0.23 ºC; Dissolved oxygen - NSE: 0.76±0.30; RMSE: 1.87±0.72 mg/L). The sediment diagenesis model exhibited similar performance metrics (Water temperature - NSE: 0.95 ± 0.18; RMSE: 1.13 ± 0.28 °C; Dissolved oxygen - NSE: 0.71 ± 0.14; RMSE: 2.01 ± 0.59 mg/L). Overall, the results suggest that the diagenesis model may be better suited for capturing detailed process-based dynamics over extended timeframes, whereas simpler models, such as the Hybrid model, are more appropriate for short- to medium-term applications or situations with limited data availability. Hopefully, the results of this study will help improve water management strategies by supporting more informed model selection tailored to the temporal scope and data constraints of reservoir monitoring programs.

## 1 Introduction

Modeling water quality plays a crucial role in managing lakes and reservoirs, providing essential insights into the dynamics of nutrients, organic matter, and phytoplankton within aquatic systems (Abbaspour et al., 2015; Whitehead et al., 2009). These models simulate the physical, chemical, and biological processes that influence water quality, with examples including widely-used tools like CE-QUAL-W2 (Wells, 2021), MIKE21 (Chapman, 1996), and DYRESM (Hamilton and Schladow, 1997). The value of such modeling lies in its capacity to aid researchers and policymakers in understanding the complex interactions between various factors that impact the ecological health of water bodies (Varis et al., 1994; Loucks and Beek, 2017). However, the intricacy of these systems, combined with the substantial data requirements, often presents significant challenges for those developing and applying water quality models. Effective inflow data characterization (quantity and

quality) is hard to obtain, both for major river branches and small tributaries, as is waterbody sediment characterization related to carbon and nutrients due to the significant cost associated with the sampling and laboratorial analysis process and the fact that water management stakeholders are still more focused on the classification of waterbody water quality rather than the collection of water quality forcing data. The absence of sediment initial particulate organic carbon (POC), particulate organic nitrogen (PON) and particulate organic phosphorus (POP) data can be decisive to the overall performance of a water quality model, in essence generating an imbalance right from the start of the simulation with regard to the sediment concentration of POC, PON and POP, which then has a considerable impact on the SOD and, consequently, the waterbody dissolved oxygen (DO). When calibrating the model, water quality modelers therefore need to plug this gap by evaluating the model performance considering: i) different initial sediment oxygen demand (SOD) where a zero-order model is applied, ii) different POC, PON and POP values where a predictive diagenesis model is considered.

The main challenge with these modeling approaches is that the sources of DO depletion—such as the inflow of organic matter or algal mortality—can significantly influence DO dynamics, and these sources must be well characterized to ensure accurate predictions. While the baseline model can reproduce observed DO profiles with reasonable accuracy, its predictive reliability may be compromised if key DO sinks and sources are not well defined.

For example, the model's response to a reduction in external phosphorus loading is influenced by internal phosphorus release from sediments during anoxic periods. In CE-QUAL-W2, when a zero-order SOD model is used, the anoxic release of phosphate ($PO_4$) is modeled as a linear function of SOD: SOD [g $O_2$/m²day] × $PO_4$ release rate [g P/g $O_2$]. Thus, any error in the estimation of SOD will directly affect the predicted internal phosphorus loading, and by extension, the overall phosphorus balance in the waterbody. In contrast, when using the predictive sediment diagenesis model, internal phosphorus loading depends on the organic and nutrient inputs from particulate matter in the water column and the sediment's biogeochemical response, which is highly influenced by the initial value of particulate organic carbon (POC). As a result, this approach introduces additional uncertainty when key particulate components are not adequately measured or constrained in both the water column and sediments. Calibrating other constituents, such as orthophosphate (P-$PO_4$), can help reduce uncertainty. P-$PO_4$ is released from sediments under anaerobic conditions, and its calibration can enhance the accuracy of DO modeling. Still, this release is influenced by multiple factors, including the initial sediment P-$PO_4$ concentration and the release rate (in the zero-order model), or the mineralization of POP (in the diagenesis model). In both cases, significant uncertainty remains without observed data for POC, PON, and POP in both the water column and sediments. Of these, POC has the most significant influence on SOD, making access to sediment POC data essential for improving model accuracy, even when PON and POP measurements are lacking. The CE-QUAL-W2 model has been widely used to simulate various water bodies and water quality scenarios, including reservoir physical and biochemical dynamics in response to warming projections (Mi et al., 2020; Mi et al., 2023). This model has also been used to predict DO in a number of water bodies worldwide, although the SOD has always been modeled with a zero-order and/or 1-order model (e.g. Park et al., 2014; Zouabi-Aloui et al., 2015; Terry et al., 2017; Sadeghian et al., 2018; Lindenschmidt et al., 2019). The bibliographic research conducted before and during this study suggests that the CE-QUAL-W2 sediment diagenesis model has not been applied to

any waterbodies other than the Wahiawa Reservoir in central Oahu (Berger and Wells, 2014). Moreover, no scientific publications on the evaluation of this model in other contexts have been identified, further highlighting the importance of the primary motivation for this study, namely, to evaluate the performance of the CE-QUAL-W2 model with its new sediment diagenesis component. This study benefited from having access to observed reservoir sediment total organic carbon (TOC) values, which are rare. Although, in theory, these values are typically higher than particulate organic carbon (POC) values, they provided an excellent starting point for this study. The methodological approach was, therefore, defined to evaluate the performance of the CE-QUAL-W2 model considering the new state-of the art sediment diagenesis model in modeling a reservoir, DO, Total Phosphorus (Total P); Total Nitrogen (Total N), Biochemical Oxygen Demand (BOD$_5$), Chlorophyll-a (Chl-a) and SOD.

To achieve this, the water quality of a highly productive reservoir was simulated over a six-year period (2016–2021) using the CE-QUAL-W2 v4.5 model. The simulation incorporated a Zero-order sediment model, a First-order model, a Hybrid model combining both approaches, and a sediment diagenesis model. The Zero-order, First-order, and Hybrid models were included to provide alternative representations of sediment oxygen demand, enabling comparative analysis and supporting the calibration and evaluation of the more complex sediment diagenesis model. In the case of water temperature and DO, the modeling results were compared with 35 water column profiles observed near the dam. The remaining parameters were calibrated using time series datasets collected at multiple depths, with six annual values available for each parameter. A sensitivity analysis was performed to evaluate the reservoir water quality response, namely DO, to the variation of POC, PON and POP concentration in the reservoir sediments. The results of this study will hopefully prove useful by helping to improve lake and reservoir water quality modeling and, therefore, the water management process from a practical perspective.

## 2 Methods

### 2.1 Site Location and Main Characteristics

Portugal experiences a temperate maritime climate characterized by a wet, cool season and a dry summer. Despite most of the precipitation occurring during the winter months, there is significant inter-annual variability. Precipitation patterns are spatially and temporally heterogeneous, with annual maxima exceeding 2500 mm in the rugged highlands of the northwest, while the low-lying plains of the southeast receive around 400 mm per year (Cardoso et al., 2013; Soares et al., 2015) (Fig.1). The Torrão dam, located in the northern region of mainland Portugal in the Tâmega River, is a significant hydraulic structure designed for multiple purposes, including water supply, irrigation, and hydroelectric power generation. The reservoir has a substantial storage capacity, contributing to regional water management and flood control. This infrastructure plays a crucial role in the socio-economic development of the region, balancing resource management and environmental preservation. However, it is also important to note that the reservoir was classified as eutrophic for all the simulated years, a condition that can lead to persistent water quality issues.

**Table 1: Main features of Torrão dam and reservoir**

| Full supply volume (hm³) | Mean inflow (m³/s) | Average annual inflow (local basin) | Active storage volume (hm³) | Surface area at FLV (km²) | Structural height (m) | Max depth (m) | Turbine number/power | Hydraulic residence time (days) | Watershed area (km²) | Trophic state (2016 - 2021)[1] |
|---|---|---|---|---|---|---|---|---|---|---|
| 123.9 | 76.98 | 2 147 | 40.4 | 6.5 | 70 | 58 | 2 reversible pump-turbines/146 (MW) | 13.59 | 3 252 | Eutrophic |

[1]   Classification according to OECD Trophic State limits (OECD, 1982)

## 2.2. CE-QUAL-W2 v4.5 model

This study employed the latest version of CE-QUAL-W2 (Version 4.5), a model originally developed in 1975 by the US Army Corps of Engineers and written in Fortran. Since its inception, the model has undergone regular updates and enhancements, primarily by researchers at Portland State University (Cole and Wells, 2006). CE-QUAL-W2 is a two-

dimensional, laterally averaged hydrodynamic and water quality model capable of simulating free surface elevation, hydrostatic pressure, density, horizontal and vertical velocities, as well as constituent concentrations. The model uses the finite difference method to solve key equations, including mean transverse momentum in the x- and z-directions, the continuity equation, state equations, and water surface elevation equations (Tavera-Quiroz et al., 2024; Wells, 2021). This model represents SOD through four distinct approaches: (i) a user-defined zero-order formulation that is decoupled from the

water column, (ii) a simple predictive first-order model, (iii) a hybrid approach combining the zero- and first-order methods, and (iv) a comprehensive sediment diagenesis model. The zero-order model is not a predictive approach, as, other than variations resulting from the temperature dependence of the decay rate, the rates remain constant over time (Wells, 2021). Additionally, under anoxic conditions in the water column, SOD is disabled in the model. The first-order sediment model does not function as a full sediment diagenesis model, as it lacks the capability to track the fate of organic nutrients delivered

to the sediments, their breakdown, and the release of byproducts into the water column under low-oxygen conditions. However, it does represent the deposition of particulate organic matter and dead algal biomass, along with the resulting oxygen demand imposed on the water column. By including this first-order sediment process, the model becomes sensitive to increased organic loading to the sediment, which in turn influences sediment oxygen demand. A combination of the zero and first order model can be considered where organic materials accumulate and decay in the sediments under aerobic

conditions and are released based on the SOD zero-order decay rate under anaerobic conditions. In contrast, the sediment diagenesis model simulates kinetic processes occurring within the sediment and at the sediment–water interface. This module originally developed for application in oil sand pit lakes, has been adapted for application in other aquatic environments and integrated into version 4.0 (Vandenberg et al., 2015). The conceptual framework of the model has been elaborated in works by Prakash et al. (2014), Berger and Wells (2014), and Vandenberg et al. (2015). It is important to note

that significant enhancements to the sediment diagenesis module were introduced in version 4.5 of the model, as detailed in

the User Manual (Wells, 2021). These improvements mark a substantial advancement over the initial version 4, which was more limited in its capabilities. The CE-QUAL-W2 model has demonstrated its utility in simulating hydrodynamic and ecological processes—such as stratification, internal waves, DO dynamics, and phytoplankton blooms—in lakes and reservoirs worldwide (Zhang et al., 2015; Chuo et al., 2019; Kobler et al., 2018; Uhlmann, 2017; Terry et al., 2017; Mi et al., 2020). Additional details about the model's structure, algorithms, and historical applications can be found in the user manual (Wells, 2021).

### 2.2.1 Model Setup

The bathymetry of the Torrão reservoir was initially defined using a Digital Elevation Model (DEM) provided by Energies of Portugal, S.A. (EDP) and structured according to the methodology outlined in Wells (2021). The reservoir comprises one main branch (the Tâmega River), three tributaries and one distributed tributary (Fig. 1). Tributaries 1 and 2 are depicted in Fig 1. Tributary 3 represents the inflow from the Douro River into the pump-back system of the Torrão Reservoir. The bathymetric map includes 27 segments, each measuring 1000 meters in length, and a maximum number of 58 layers, each with a depth of 1 meter. Following this preliminary step, the reservoir boundary conditions (including water quality, hydrology, meteorology, and sediment characterization) were defined according to the methods described in Section 2.4. Due to the lack of available information, the model structure only includes a single algae group (Diatoms).

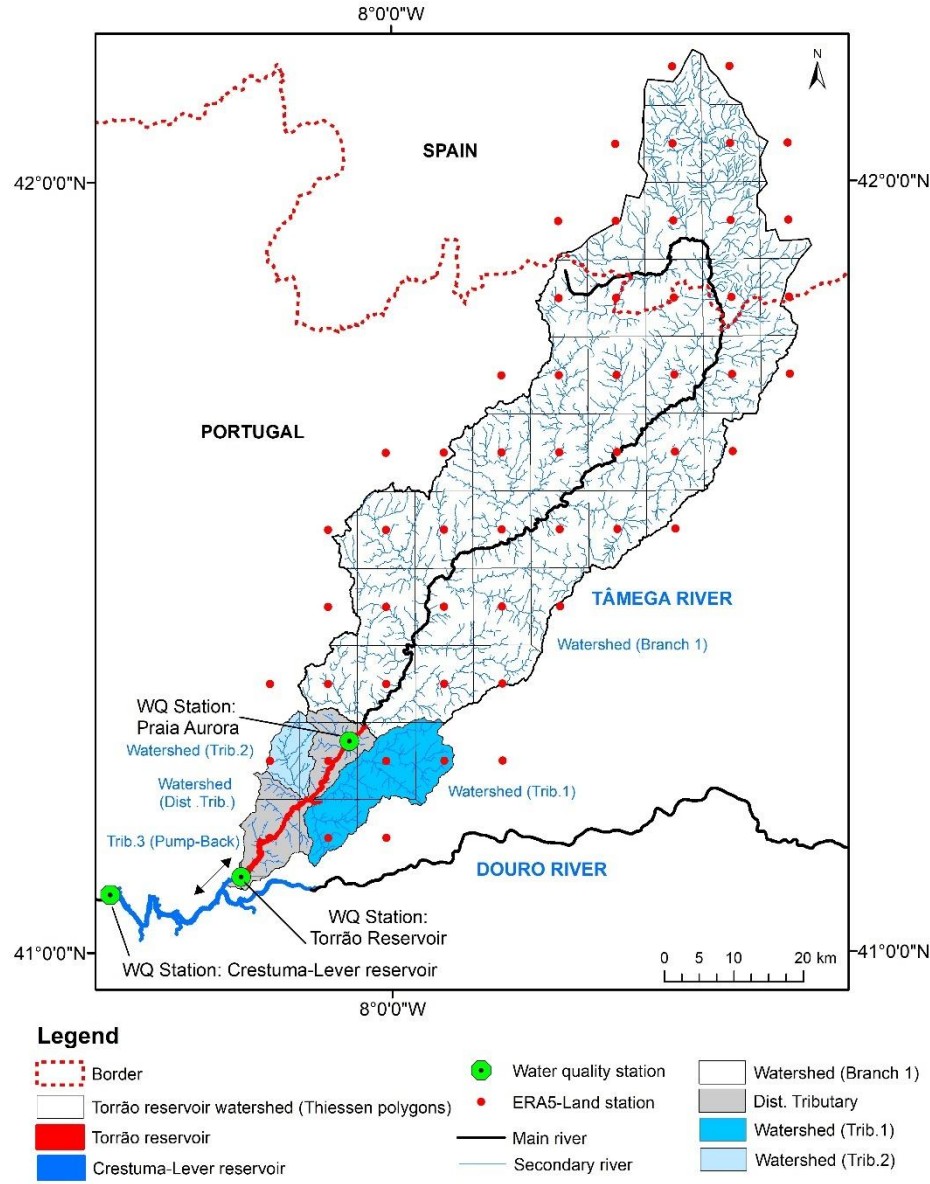

**Figure 1: Torrão reservoir watershed. Thiessen polygons. Water quality stations**

### 2.3 Modeling approach

To thoroughly evaluate the capability of CE-QUAL-W2 in modeling dissolved oxygen using the sediment diagenesis module, the four available SOD modeling approaches were considered: Zero-order model; First-order model; Zero/First-order model (Hybrid model) and the sediment diagenesis model (SG model). The models were calibrated for the 2016–2021

period (see Section 2.5). During the results analysis, the performance metrics obtained during each model's calibration process were compared, along with the SOD values across the bottom layers of each model. A sensitivity analysis was conducted following calibration to evaluate each model's response: a) to varying POC, PON, and POP values in the case of the SG model; b) to different SOD values in the Zero-order and Hybrid models and c) to varying the initial first order sediment concentration in the case of the First-order model. Section 2.6 details the methodological approach used for the sensitivity analysis. To assess the sensitivity of each model to reductions in external organic matter (OM) and phosphorus ($PO_4$-P) inputs, two separate scenario analyses were conducted. The first scenario involved an 80% reduction in OM inflow load, while the second applied an 80% reduction in both OM and $PO_4$-P inflow loads. These reductions were implemented specifically in the main reservoir branch (Branch 1 – Tâmega River), where the majority of nutrient and organic inputs occur. Each sediment model—SD, Zero-order, First-order, and Hybrid—was run under baseline conditions and under both reduction scenarios. The impact on DO dynamics was evaluated using time series of depth- and segment-averaged DO concentrations. Each model—SD, Zero-order, First-order, and Hybrid—was run under baseline conditions and then under this reduced-loading scenario. The evaluation of model performance, along with the results of the sensitivity analysis, provided deeper insights into simulating SOD dynamics using the sediment diagenesis approach in comparison to the other SOD formulations.

**2.4 Model Forcing Datasets**

The meteorological data used to drive the model, including hourly air temperature, dew point, solar radiation, cloud cover, and wind characteristics, were sourced from ERA5-Land, a high-resolution reanalysis dataset optimized for land applications. Although no on-site meteorological stations are available in the study area for direct validation, studies by Almeida and Coelho (2023b) and Barbosa et al. (2022) have demonstrated a strong correlation between ERA5-Land air temperature data and observed measurements at regional scales, supporting the reliability of this dataset for our modeling purposes. Furthermore, the accuracy of water temperature predictions in our simulations indicates that the meteorological forcing was well represented, confirming the suitability of ERA5-Land data for driving the model. Reservoir data, such as daily inflow/outflow, water levels, and water quality, covering the years 2016–2021, were provided by EDP. Water quality data specific to Branch 1 originated from the Praia Aurora Station, accessed via the Portuguese National Water Resources Information System (SNIRH, 2024). With only 21 recorded measurements for Branch 1 during this period, three modeling methods were employed to address the 99.04% of missing data. The variables include: water temperature; DO; Total P; Ammonium (N-NH$_4$); Nitrate+Nitrite (N-NO$_X$); BOD$_5$; Chl-a; Alkalinity; Conductivity and Total Suspended Solids (SST). The first method employed regression models implemented through the LOADEST package (Runkel et al., 2004) developed by the U.S. Geological Survey. The second method utilized the Extreme Gradient Boosting (XGBoost) machine learning algorithm, implemented using the Chen and Guestrin (2016) open-source library, a method proven effective in various environmental studies (Feigl et al., 2021; Adedeji et al., 2022; Xu et al., 2022). For additional details on the algorithm, refer to Almeida and Coelho (2023b). The third approach relied on Support Vector Regression (SVR), implemented via the scikit-

learn library (Pedregosa et al., 2011), which has also demonstrated strong performance in environmental modeling applications (Adedeji et al., 2022; Ji and Lu, 2018). For machine learning approaches, datasets were split into training (80%) and testing (20%) sets. Hyperparameters for these models were optimized using the Tree-structured Parzen Estimators (TPE) algorithm, executed with the Hyperopt library (Bergstra et al., 2013) and 100 iterations. The Nash-Sutcliffe Efficiency (NSE) was used to determine the best model. Table A1 describes the input features of each model. Correlations derived from Branch 1 informed data extrapolation to other tributaries using flow as the predictor. Observed data for Tributary 3 was retrieved from the Crestuma-Lever reservoir monitoring station.

Water quality variables used for model inputs included water temperature, DO, orthophosphates (P-PO$_4$), N-NH$_4$, N-NOx, labile and refractory dissolved and particulate organic matter (LDOM, RDOM, LPOM, RPOM), alkalinity, inorganic suspended solids (ISS), total dissolved solids (TDS), total inorganic carbon (TIC), and algal biomass (diatoms). For non-monitored variables, estimations were made based on available data: i) P-PO$_4$: Derived from total phosphorus, assuming inorganic phosphorus represents 70% of the total; ii) Organic matter: BOD$_5$ was converted to organic matter using a stoichiometric ratio of 1.4 g O$_2$/1.0 g organic matter, with 60% assumed refractory and 40% labile; iii) ISS: Estimated as 97.4% of TSS; iv) TDS: Calculated from electrical conductivity (Eq.1); v) TIC: Estimated from alkalinity (Eq. 2); vi) Algae biomass: Chl-a was converted to biomass using the following ratio: Algal Biomass (mg/L)/ Chl-a (µg/L) = 0.05

$$TDS\ (mg/L) = 0.65 \times Electrical\ Conductivity(\mu S/cm) \tag{1}$$
$$TIC\ (mg/L) = 0.2782 \times Alkalinity\ (mg/L)0.9706 \tag{2}$$

This equation was derived from the relationship between TIC and alkalinity values observed in four reservoirs within the United States, utilizing a dataset comprising 55232 value pairs available in the CE-QUAL-W2 v4.5 model examples (Wells, 2021). The analysis achieved an $R^2$ value of 0.99. Figure 2 illustrates the locations of the five sediment sampling sites used to define the SD model baseline run. The spatial distribution of the sediment samples depicted in the figure were linked to specific reservoir segments to characterize the initial sediment content of POC, PON, and POP, as detailed in Table 2. Sediment values were assigned as follows: site A to segments 25–28, site B to segments 20–24, site C to segments 16–19, site D to segments 11–15, and site E to segments 2–10. Several assumptions were made to establish the sediment characterization: i) A sediment density of 960 kg/m³ (density of dried sediment with air in the pore space) (Minear, 2007) was applied to convert sample values from mg/kg to mg/L; ii) POP values were set at 25 mg/L, based on established literature benchmarks (Wells, 2021); iii) The TN value observed at site B was used to characterize sites C, D, and E; iv) TOC and TN were assumed to exist entirely in particulate form, represented as particulate organic carbon and nitrogen. This approach ensured a consistent and representative characterization of sediment properties across the reservoir segments.

**Table 2: Torrão Reservoir sediment chemical characterization obtained for each sampling site.**

| Observed values | | | |
|---|---|---|---|
| **Sampling site** | **TOC, mg/kg** | **TN, mg/kg** | **POP, mg/kg** |
| A | 25 000 | 6020 | - |
| B | 20 900 | 5990 | - |
| C | 22 300 | - | - |
| D | 20 100 | - | - |
| E | 5 600 | - | - |
| Final values included in the sediment diagenesis model | | | |
| | **POC, mg/L** | **PON, mg/L** | **POP, mg/L** |
| A | 24 000 | 5779 | 25 |
| B | 20 064 | 5750 | 25 |
| C | 21 408 | 5750 | 25 |
| D | 19 296 | 5750 | 25 |
| E | 5376 | 5750 | 25 |

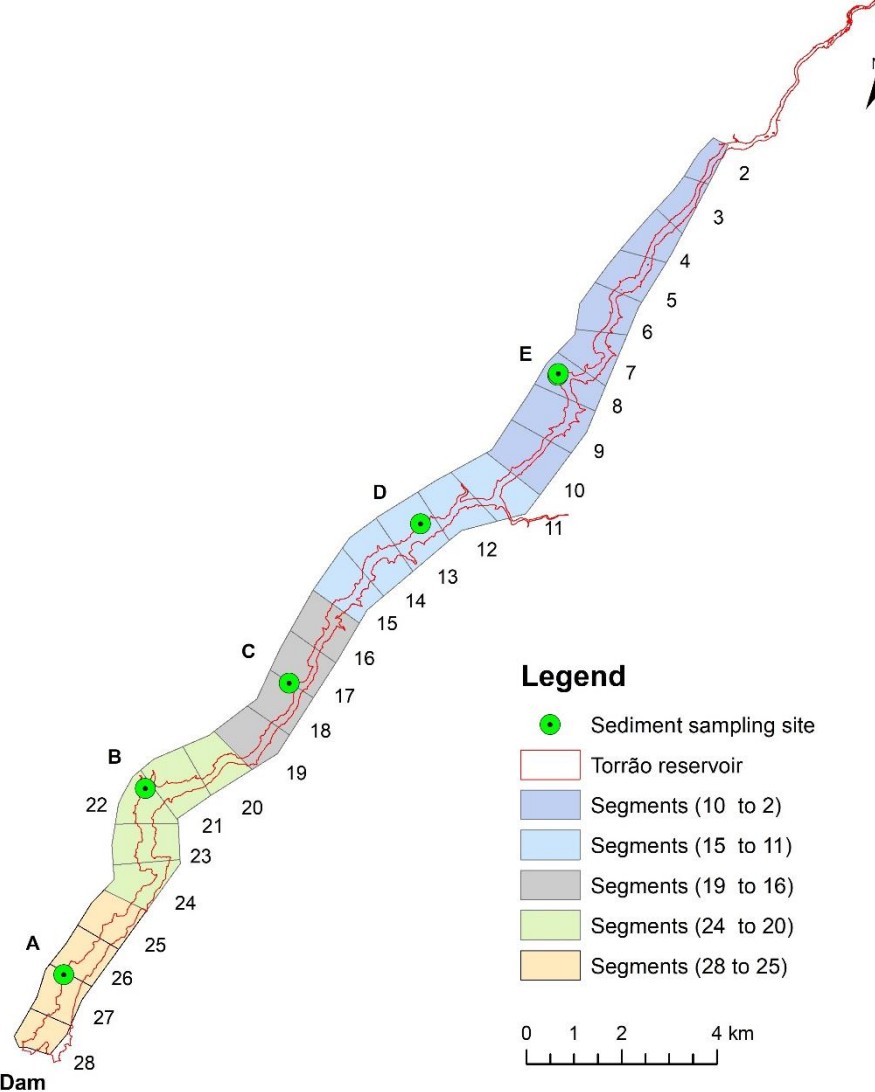

**Figure 2: Sediment sampling sites. CE-QUAL-W2 model segments**

**2.5 Water Quality Model (CE-QUAL-W2) Calibration**

The simulation period considered for this study spanned 2016 to 2021. This period was selected due to the availability of flow and water quality data. The trial-and-error technique was applied to calibrate the model for the simulation period, considering the default calibration parameters described in Wells, 2021. The error between observed and predicted values of six state variables was evaluated with five different metrics (see section 2.7). The observed data included six annual values 225 for water temperature, DO, TP, TN, $BOD_5$, and Chl-a. These time series were obtained from: (a) an integrated sample

between the reservoir surface and a depth of 5.8 meters, (b) a depth of 23 meters, and (c) a depth of 43.7 meters. In addition, 35 water temperature and DO profiles—six per year from 2016 to 2021—were also included. These profiles were observed 300 meters upstream from the Torrão Dam. Details on the models' initial conditions, parameters, constants, and forcing datasets can be found in Almeida and Coelho (2025) and in Tables A2 to A8. The models were calibrated by adjusting their parameters to improve the fit between the model output and observed data. Please refer to Wells (2021) for a detailed account of the model calibration parameters and default values. Water temperature was the first constituent to be calibrated. The wind sheltering coefficient (WSC) was manually adjusted to achieve the best fit between the modeled and observed water temperature profiles, resulting in a final value of 1. A value of 1 implies that the WSC has no effect over the wind velocity forcing the model. The zero-order model for SOD was then manually adjusted to improve DO predictions based on 35 DO profiles. The optimal result was achieved with a zero-order SOD value of 2.5 g $O_2$/m²day. Following this calibration, the phosphorus sediment release rate (PO4R) in the zero-order model was modified from its default value of 0.015 to 0.001. The same process was applied to the Hybrid model, where the best results were achieved using a PO4R of 0.001 and a zero-order SOD value of 1.0 g $O_2$/m²day. In the first-order model, the $PO_4R$ parameter was adjusted to 0.001, and the initial concentration of first-order sediment was set to 0.5 g/m². All other parameters were kept at their relevant default values and the default settings for the sediment diagenesis model were also maintained. The observed data included water temperature, DO), TP, TN, $BOD_5$, and Chl-a.

## 2.6 Sensitivity analysis

A sensitivity analysis was conducted after the calibration process to evaluate the model's response:

i)   Different initial sediment values for POC, PON, and POP were used in the SD model (Table 3). It is important to note that for each of the 24 runs described in Table 2, only the corresponding parameter was modified, while the other two parameters retained their default values shown in Table 3. The number of runs varying the PON and POP values is higher than the number of runs considered for POC, with 6 versus 9 runs, respectively. This adjustment was necessary to achieve a minimal RMSE in the predictions of dissolved oxygen in the water column;

ii)  Different zero-order SOD values for the Zero-order model (0.5, 1.0, 1.5, 2.0, 2.5 and 3.0 g $O_2$/m².day);

iii) Different initial first order sediment concentration (ISC) for the First-order model (0.0, 0.5, 1.0, 1.5, 2.0, 2.5 and 3.0 g/m²);

iv)  Different zero-order SOD values for the Hybrid model (0.5, 1.0, 1.5, 2.0, 2.5 and 3.0 g $O_2$/m².day.

In the results analysis for each run and for both scenarios (i) and (ii), the prediction error for DO was compared with the SOD values derived from each model. Specifically, runs 5, 8, and 20 were forced with the POC, PON, and POP values defined in the SD model baseline run.

**Table 3. TOC, PON, and POP initial sediment values for the SD model sensitivity analyses.**

| Parameter | Sampling site/Model segments | | | | | SD model run |
| | A | B | C | D | E | |
| | 28-25 | 24-20 | 19-16 | 15-11 | 10-2 | |
|---|---|---|---|---|---|---|
| POC | 4 800 | 4 013 | 4 282 | 3 859 | 1 075 | 1 |
| | 9 600 | 8 026 | 8 563 | 7 718 | 2 150 | 2 |
| | 14 400 | 12 038 | 12 845 | 11 578 | 3 226 | 3 |
| | 19 200 | 16 051 | 17 126 | 15 437 | 4 301 | 4 |
| | 24 000 | 20 064 | 21 408 | 19 296 | 5 376 | 5 (SD model-baseline) |
| | 28 800 | 24 077 | 25 690 | 23 155 | 6 451 | 6 |
| PON | 8 669 | 8 625 | 8 625 | 8 625 | 8 625 | 7 |
| | 5 779 | 5 750 | 5 750 | 5 750 | 5 750 | 8 (SD model-baseline) |
| | 2 890 | 2 875 | 2 875 | 2 875 | 2 875 | 9 |
| | 1 445 | 1 438 | 1 438 | 1 438 | 1 438 | 10 |
| | 722 | 719 | 719 | 719 | 719 | 11 |
| | 361 | 359 | 359 | 359 | 359 | 12 |
| | 181 | 180 | 180 | 180 | 180 | 13 |
| | 90 | 90 | 90 | 90 | 90 | 14 |
| | 45 | 45 | 45 | 45 | 45 | 15 |
| POP | 128 | 128 | 128 | 128 | 128 | 16 |
| | 85 | 85 | 85 | 85 | 85 | 17 |
| | 57 | 57 | 57 | 57 | 57 | 18 |
| | 38 | 38 | 38 | 38 | 38 | 19 |
| | 25 | 25 | 25 | 25 | 25 | 20 (SD model-baseline) |
| | 13 | 13 | 13 | 13 | 13 | 21 |
| | 6 | 6 | 6 | 6 | 6 | 22 |
| | 3 | 3 | 3 | 3 | 3 | 23 |
| | 2 | 2 | 2 | 2 | 2 | 24 |

## 2.7 Metrics

The evaluation of model calibration and the analysis of quantitative differences across simulation scenarios utilized various performance metrics. These included the root mean square error (RMSE), mean absolute error (MAE), Nash-Sutcliffe efficiency (NSE) (Nash and Sutcliffe, 1970), percent bias (PBIAS), and the coefficient of determination ($R^2$). The

265 calculations were carried out using equations where $m_i$ and $o_i$ represent the simulated and observed values, respectively, and $\bar{o_i}$ the observed values mean.

$$RMSE = \sqrt{\frac{1}{N}\sum_{i=1}^{N}(m_i - o_i)^2} \tag{3}$$

$$MAE = \frac{1}{N}\sum_{i=1}^{N}|m_i - o_i| \tag{4}$$

$$NSE = 1 - \left[\frac{\sum_{i=1}^{N}(o_i - m_i)^2}{\sum_{i=1}^{N}(o_i - \bar{o}_i)^2}\right] \tag{5}$$

$$PBIAS = \frac{\sum_{i=1}^{N}(o_i - m_i)}{\sum_{i=1}^{N}(o_i)} \times 100 \tag{6}$$

$$R^2 = \frac{\sum_{i=1}^{N}(m_i - \bar{o})^2}{\sum_{i=1}^{N}(o_i - \bar{o})^2} \times 100 \tag{7}$$

## 3 Results

### 3.1 Observed Inflow Water Quality Characterization

The SVR algorithm was more effective at predicting the inflow water temperature compared to the other models. The $R^2$ and PBIAS values achieved with the SVR were 0.87, and 3.77%, respectively, indicating that the water temperature trends and average magnitudes are well described (Table A1). Additionally, the RMSE and MAE values of 2.1ºC and 1.6ºC, respectively, demonstrate an accurate approximation of the observed datasets. The SVR algorithm was also the best model in predicting DO. The $R^2$, PBIAS, RMSE, and MAE values reached, 0.91, 0.92%, 0.40 mg/L and 0.26 mg/L, respectively, indicating that the model performed well. This was not the case for the remaining parameters. In fact, the Loadest regression outperformed the other models for the remaining water quality variables. This was primarily due to the limited number of training samples. Simpler models like regressions can have lower variance (i.e., be less susceptible to overfitting) compared to SVR and XBOOST algorithms. Overall, the PBIAS obtained for NH4, N-NOx, and Chl-a (10.88%, 43.64%, and 30.00%) suggests that the average magnitude was reasonably well represented.

### 3.2 CE-QUAL-W2 calibration

Tables A2 through A8 display the most significant CE-QUAL-W2 coefficients obtained after the calibration process. The results of the calibration process for all models, are presented in Table 4 and Table A9 and illustrated in figures 3 to 6 and figures 8 and 9. The performance metrics for water temperature across the different sediment models show consistent accuracy, with NSE and $R^2$ values ranging from 0.95 to 0.96 and minimal variation across models. The RMSE and MAE for temperature also remain low, indicating reliable thermal performance regardless of the sediment model applied. In contrast, DO predictions show more variability. The Hybrid model achieved the best overall DO performance, with the highest NSE ($0.76 \pm 0.30$) and $R^2$ ($0.76 \pm 0.31$), as well as the lowest RMSE ($1.87 \pm 0.72$) and MAE ($1.22 \pm 0.55$), while maintaining a

near-zero PBIAS (-0.55 ± 11.14), indicating minimal systemic bias. The Zero-order model also performed reasonably well, with slightly lower error metrics than the SD model. The First-order model, however, showed the weakest DO performance, with a lower NSE (0.68 ± 0.22), higher RMSE (2.15 ± 0.82), and a significant negative PBIAS (-12.17 ± 15.44), suggesting an underestimation of oxygen concentrations. Overall, the results suggest that while temperature simulation is robust across all models, DO dynamics are better captured using the Hybrid or Zero-order models, with the Hybrid model offering the most balanced and accurate representation under the tested conditions. However, the differences in performance metrics for DO among the models are relatively small and often fall within overlapping standard deviations, with the exception of the First-order model, which consistently shows lower accuracy and higher bias, suggesting that while the Hybrid model offers slightly better overall performance, the improvements over the SD and Zero-order models are modest and should be interpreted with caution. In terms of nutrient dynamics, the Hybrid and Zero-order models improve TN and TP predictions relative to the SD and First-order models. The Hybrid model, for example, improves TN $R^2$ to 0.31 and TP to 0.27, although the associated biases remain significant (e.g., −18.75% for TN and +36.49% for TP). $BOD_5$ and Chl-a remain poorly simulated across all models, with $R^2$ values consistently low (≤0.06 for Chl-a and ≤0.03 for $BOD_5$), and large PBIAS values, particularly in the SD and First-order configurations. The Zero-order model slightly reduces bias in Chl-a and Total N compared to the SD model but performs poorly for TP due to a large overestimation (PBIAS = 103.43%) (Fig.4D). Notably, the SD and First-order models failed to reproduce observed phosphorus release events from sediments on 2018-09-18, 2020-09-08, and 2021-08-31 (Figures 3D and 5D). In contrast, the Hybrid model successfully captured these events by modeling phosphorus release as a linear function of SOD, providing a more realistic representation of sediment–water nutrient interactions (Fig.6D). Overall, while no model fully captures the complexity of all constituents, the Hybrid model consistently provides the most balanced and improved representation, particularly for DO and nutrient parameters.

**Table 4: Metrics between observed and predicted values for all models. Water temperature and DO metrics were obtained from 36 observed and predicted profiles.**

| Constituent | SD model (run 5 - baseline) | | | | |
|---|---|---|---|---|---|
| | NSE | R² | PBIAS | RMSE | MAE |
| Water temperature | 0.95±0.18 | 0.96±0.07 | 1.96±3.08 | 1.13±0.28 | 0.89±0.26 |
| DO | 0.71±0.14 | 0.73±0.29 | 4.43±15.06 | 2.01±0.59 | 1.38±0.46 |
| **Constituent** | **Zero-order model (zero-order SOD = 2.5 g O$_2$/m$^2$day - baseline)** | | | | |
| | NSE | R² | PBIAS | RMSE | MAE |
| Water temperature | 0.95±0.19 | 0.96±0.07 | 1.91±3.09 | 1.13±0.28 | 0.89±0.25 |
| DO | 0.73±0.20 | 0.74±0.30 | 1.75±15.87 | 1.97±0.74 | 1.29±0.57 |
| **Constituent** | **First-order model (ISC= 0.5 g/m$^2$ - baseline)** | | | | |
| | NSE | R² | PBIAS | RMSE | MAE |
| Water temperature | 0.96±0.19 | 0.96±0.08 | 1.46±2.97 | 1.09±0.23 | 0.85±0.20 |
| DO | 0.68±0.22 | 0.73±0.27 | -12.17±15.44 | 2.15±0.82 | 1.50±0.65 |
| **Constituent** | **Hybrid model (zero-order SOD= 1.0 g O$_2$/m$^2$day - baseline)** | | | | |
| | NSE | R² | PBIAS | RMSE | MAE |
| Water temperature | 0.96±0.18 | 0.96±0.08 | 1.45±2.97 | 1.09±0.23 | 0.85±0.20 |
| DO | 0.76±0.30 | 0.76±0.31 | -0.55±11.14 | 1.87±0.72 | 1.22±0.55 |

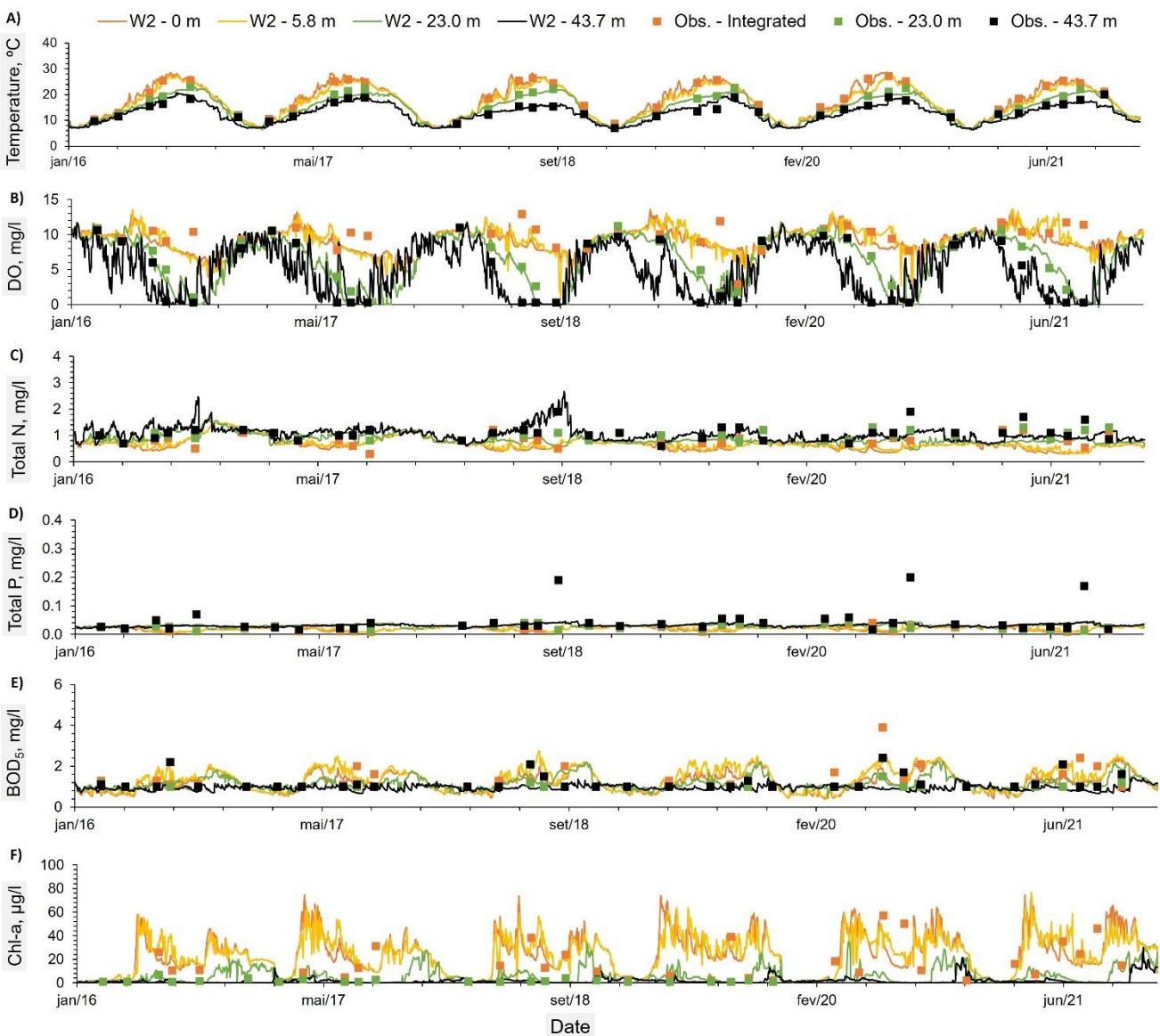

**SD model (run 5 - baseline)**

**Figure 3: Constituents observed values at three different depths: (a) an integrated sample between the reservoir surface and an average depth of 5.8 meters, (b) an average depth of 23 meters, and (c) an average depth of 43.7 meters. These observed values were compared with the predicted time series from the SD model (run 5 - baseline) (A to F) for the same depths.**

**Zero-order model (zero-order SOD = 2.5 g O₂/m²/day - baseline)**

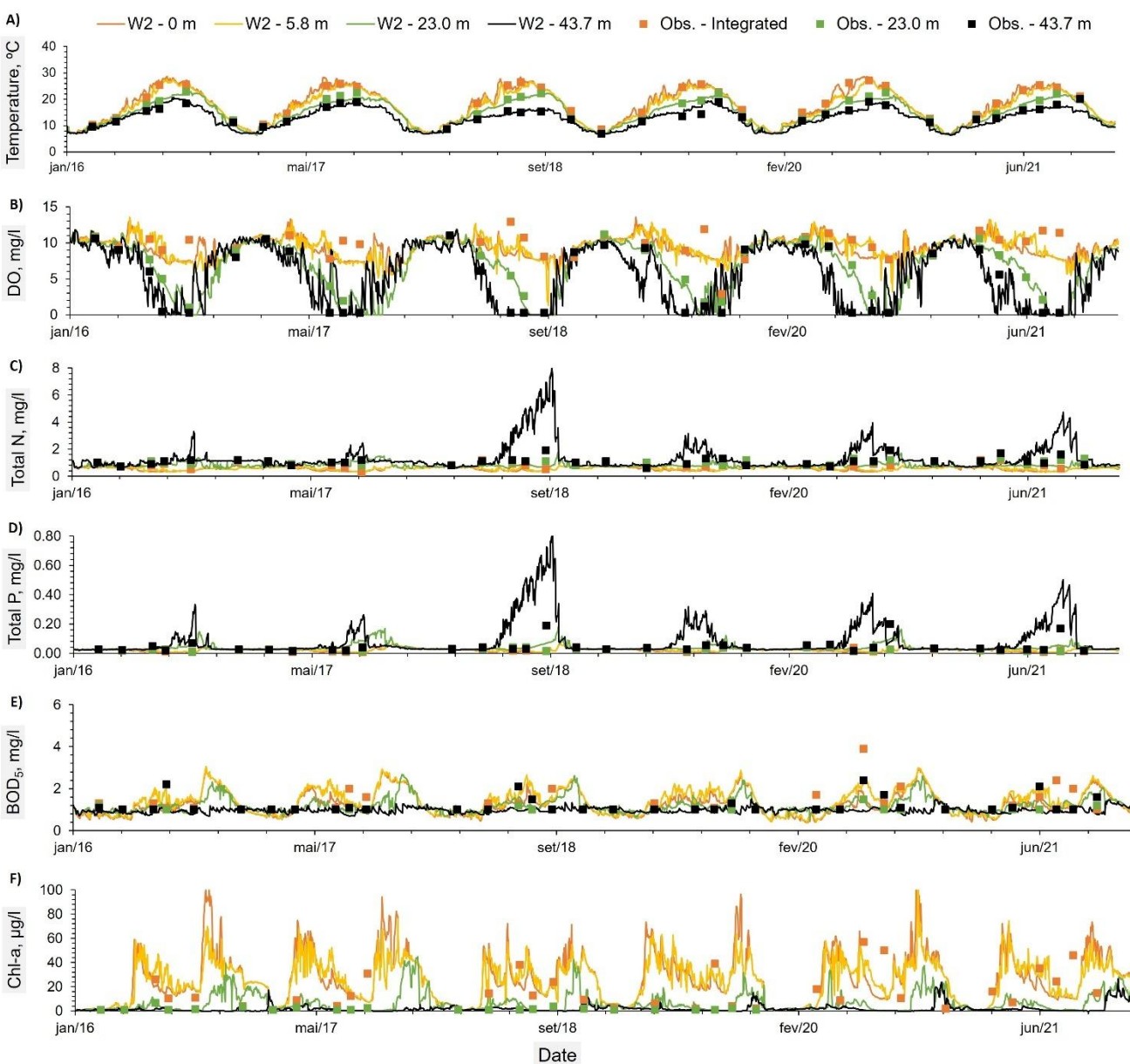

Figure 4: Constituents observed values at three different depths: (a) an integrated sample between the reservoir surface and an average depth of 5.8 meters, (b) an average depth of 23 meters, and (c) an average depth of 43.7 meters. These observed values were compared with the predicted time series from the Zero-order model (zero order SOD = 2.5 g O₂/m²day - baseline) (A to F) for the same depths.

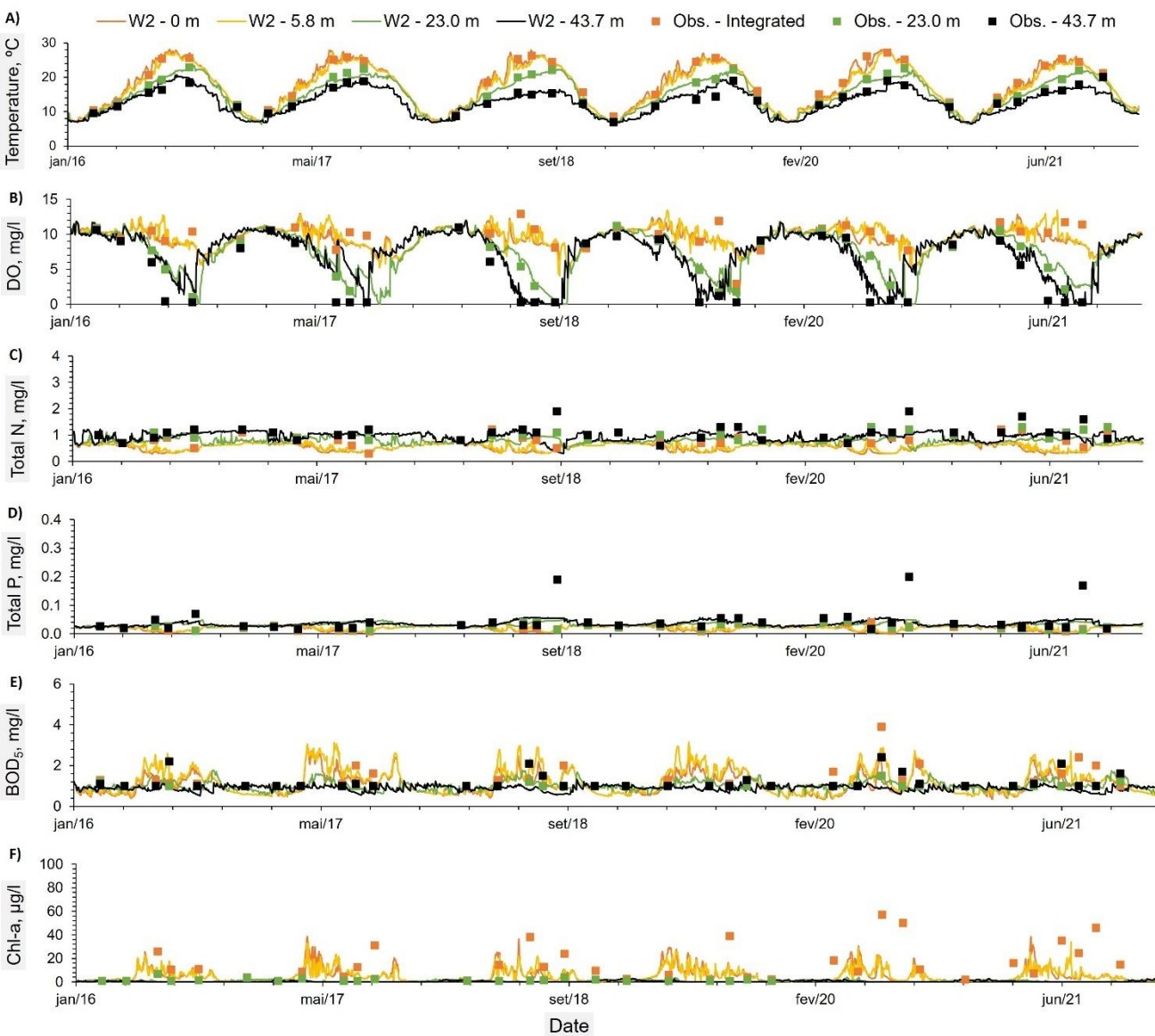

**Figure 5: Constituents observed values at three different depths: (a) an integrated sample between the reservoir surface and an average depth of 5.8 meters, (b) an average depth of 23 meters, and (c) an average depth of 43.7 meters. These observed values were compared with the predicted time series from the First-order model (ISC=0.5 g/m² - baseline) (A to F) for the same depths.**

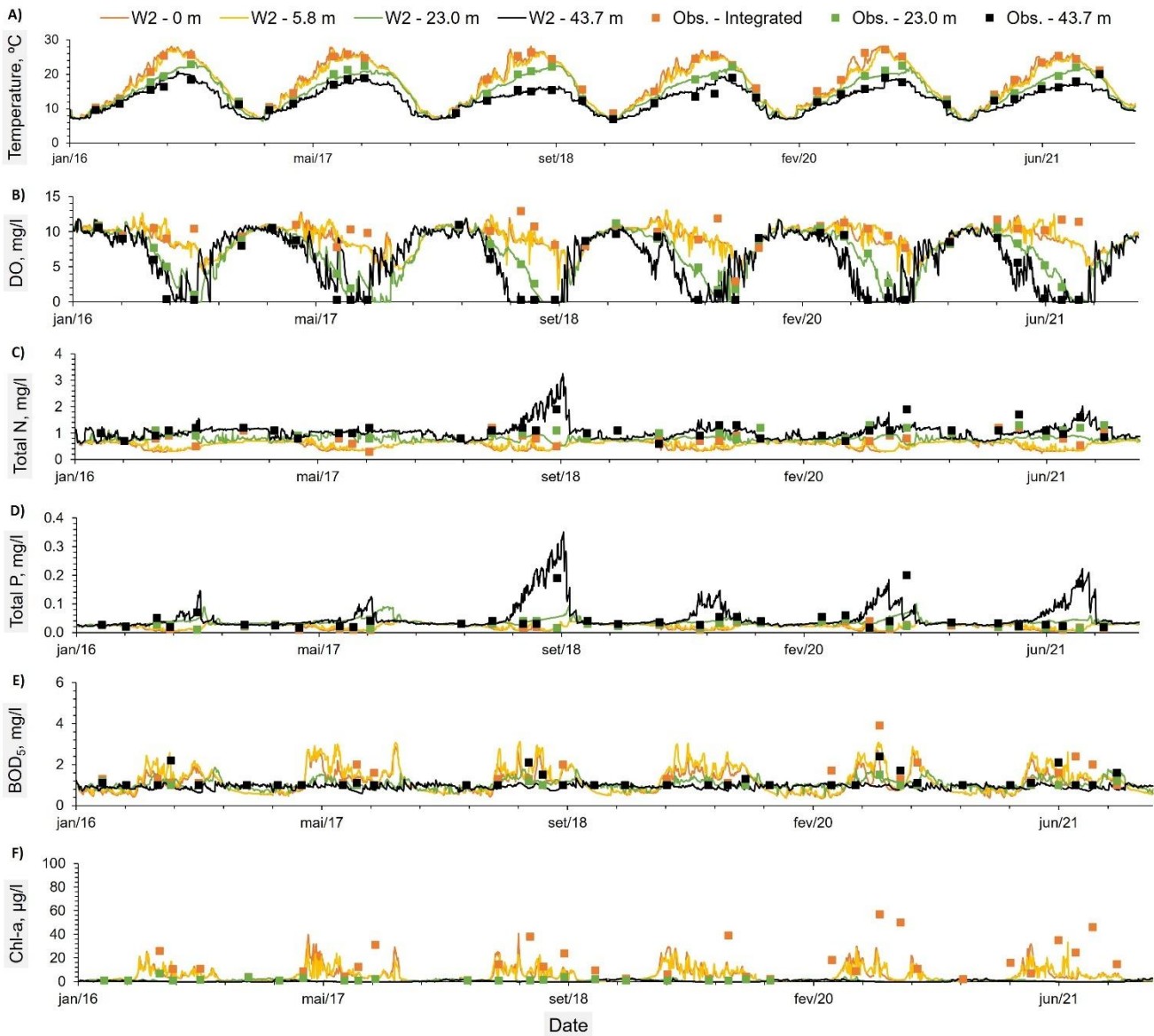

**Hybrid model (zero-order SOD = 1.0 g O₂/m²/day - baseline)**

Figure 6: Constituents observed values at three different depths: (a) an integrated sample between the reservoir surface and an average depth of 5.8 meters, (b) an average depth of 23 meters, and (c) an average depth of 43.7 meters. These observed values were compared with the predicted time series from the Hybrid model (zero order SOD= 1.0 g O₂/m²day - baseline) (A to F) for the same depths

### 3.3 Sensitivity analysis

The SOD values strongly influence the water column DO; therefore, this parameter was considered to support this analysis. Figure 7 shows the SOD values from the reservoir bottom layer, predicted by the SD model for Runs 1 to 6, compared with the RMSE (Fig7A) and the NSE (Fig7B) values obtained between the predicted water column DO profiles and the mean initial POC values (across all sites values) for each run. These results suggest that Run 4 was the best modeling solution. Considering the results obtained for Run 5 (baseline), Run 4 reduced the RMSE from 2.015 mg/L (Run 5) to 2.011 mg/L (Run 4) and increased the NSE from 0.714 (Run 5) to 0.716 (Run 4). The average SOD value in the bottom layer of the reservoir (across all model segments) decreased from 1.162 g $O_2$/m²day (Run 5) to 1.071 g $O_2$/m²day (Run 4). Although the reduction is modest and had only a minor effect on the DO profile predictions (Fig. 9), it suggests that the initial POC values used in Run 5 were likely overestimated. This outcome aligns with the assumption made in Run 5, where all observed TOC was considered to exist entirely as POC. In contrast, Run 4 was characterized using a lower average sediment concentration. Specifically, the mean value used in Run 4 (14170 mg/L) represents approximately 80% of the TOC value used in Run 5 (17712 mg/L), which was derived from observed TOC measurements (see Table 3). This comparison suggests that a more realistic estimate is that about 80% of the total organic carbon exists in particulate form, with the remainder composed of dissolved organic carbon. Run 4 and Run 5 show negligible differences in the predicted water temperature and DO profiles (Fig. 8 and 9). Table A10 presents the performance metrics for water temperature, DO, TN, TP, BOD$_5$, and Chl-a obtained for Run 4. While this run improved the DO simulation in the reservoir, results for the other constituents remained very similar to those of Run 5 (baseline). Overall, the water temperature profiles are very well captured by all models (Fig. 8), reflecting their robustness in simulating thermal dynamics. In contrast, DO profiles are more complex and challenging to model due to their sensitivity to multiple interacting processes. Nevertheless, the models were able to capture the main seasonal and vertical trends in DO concentrations, including stratification patterns and general oxygen depletion in bottom layers during warmer months (Fig.9).

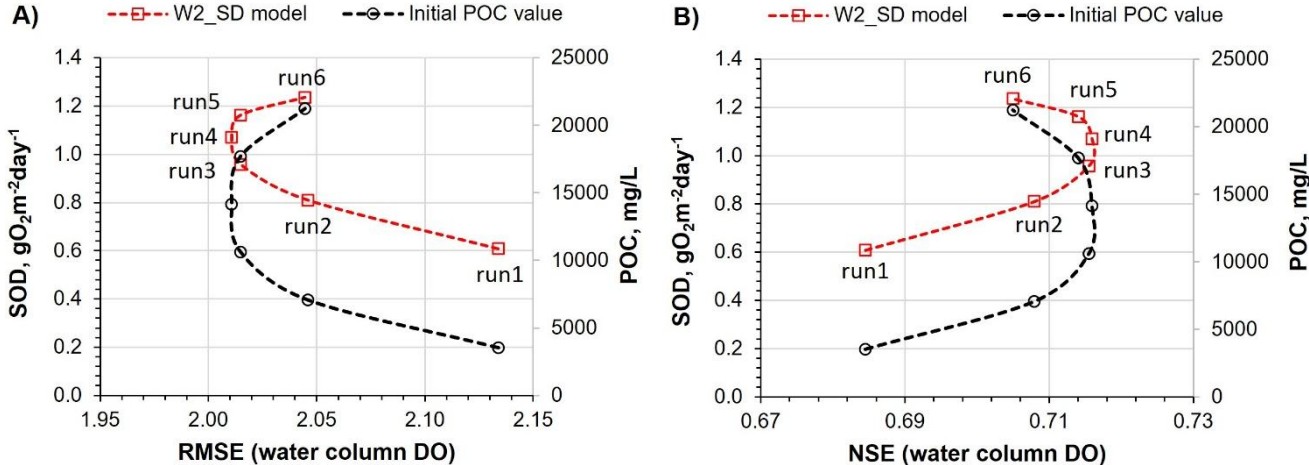

**Figure 7: A) SOD values from the reservoir bottom layer, predicted by the SD_model for Runs 1 to 6, compared with the RMSE obtained between the predicted water column DO profiles and the mean initial POC values (across all sites values) for each run of the SD_model. B) Similar to A but considering the NSE metric.**

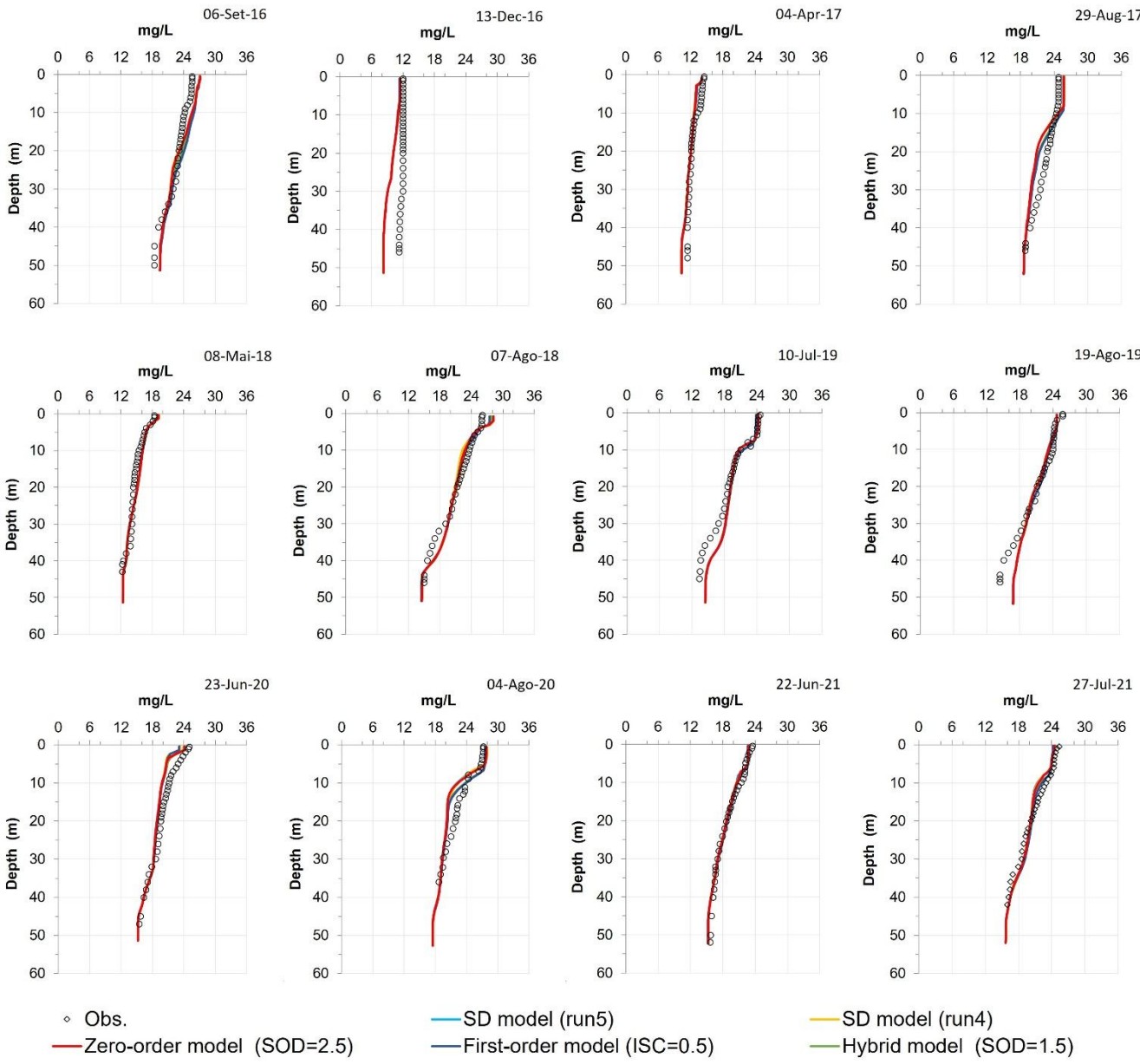

**Figure 8: Observed water temperature profiles (300 m from the dam) compared to predicted profiles using the SD model (Run 4) and (Run 5 - baseline), Zero-order model (zero-order SOD = 2.5 g O$_2$/m$^2$day - baseline); First-order model (ISC= 0.5 g/m$^2$ - baseline) and the Hybrid model (zero order SOD= 1.0 g O$_2$/m$^2$day - baseline).**

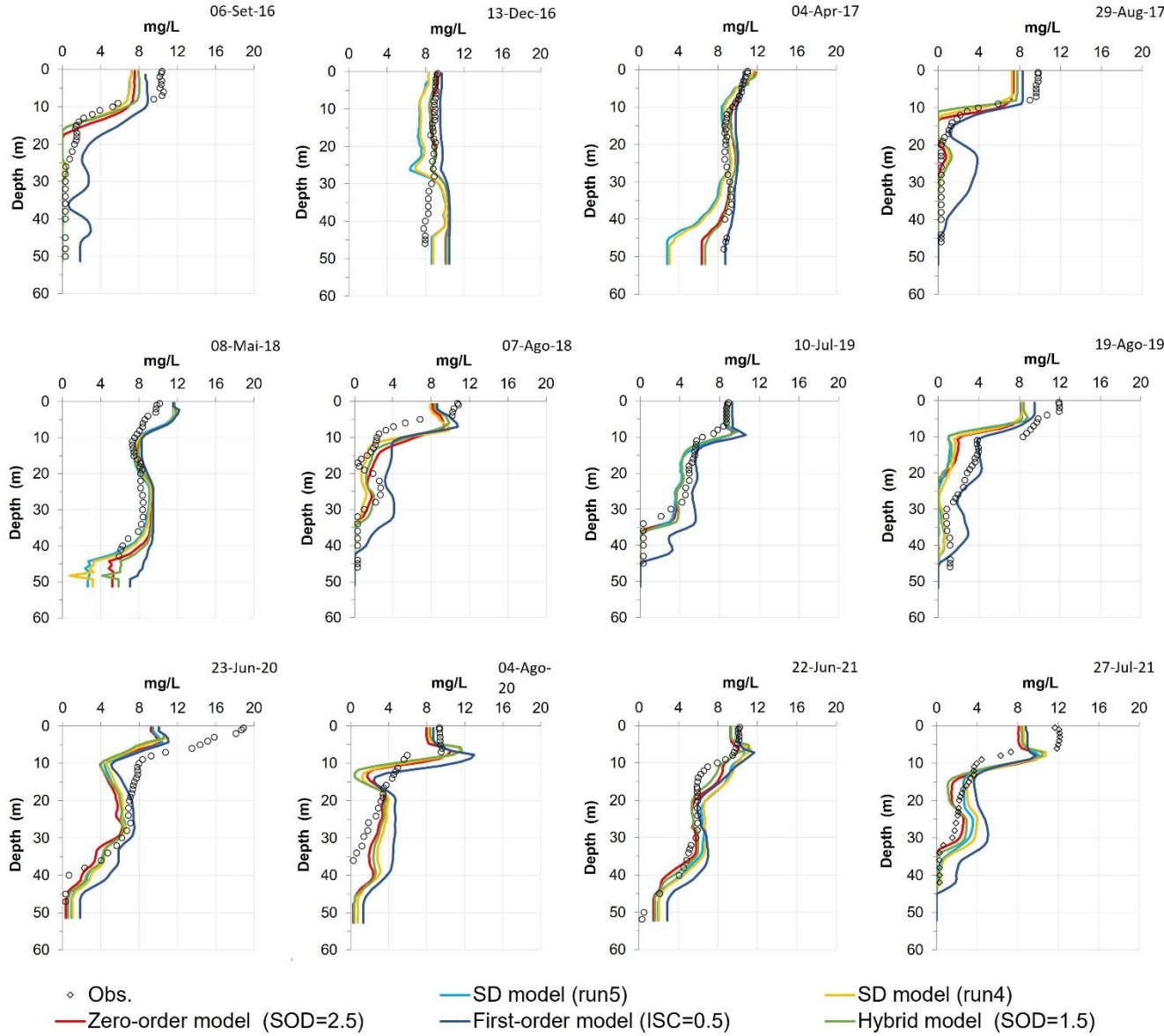

**Figure 9: Observed DO profiles (300 m from the dam) compared to predicted profiles using the SD model (Run 4) and (Run 5 - baseline), Zero-order model (zero-order SOD = 2.5 g O₂/m²day - baseline); First-order model (ISC= 0.5 g/m² - baseline) and the Hybrid model (zero order SOD= 1.0 g O₂/m²day - baseline).**

The sensitivity analysis also involved varying the initial values of PON and POP for each run. The results indicate that mean reservoir SOD values remained nearly constant, as depicted in Fig. 10, suggesting that the SD model was not significantly affected by variations in the initial PON and POP values in the sediments. However, in Runs 7, 8, and 9, where PON values were higher, there was a significant increase in the release of N-NH₄ and N-NOₓ from the reservoir sediments, leading to an

impact on water column DO. This is evidenced by the notable increase in RMSE and the reduction of NSE values, as shown in Fig. 10A and 10B.

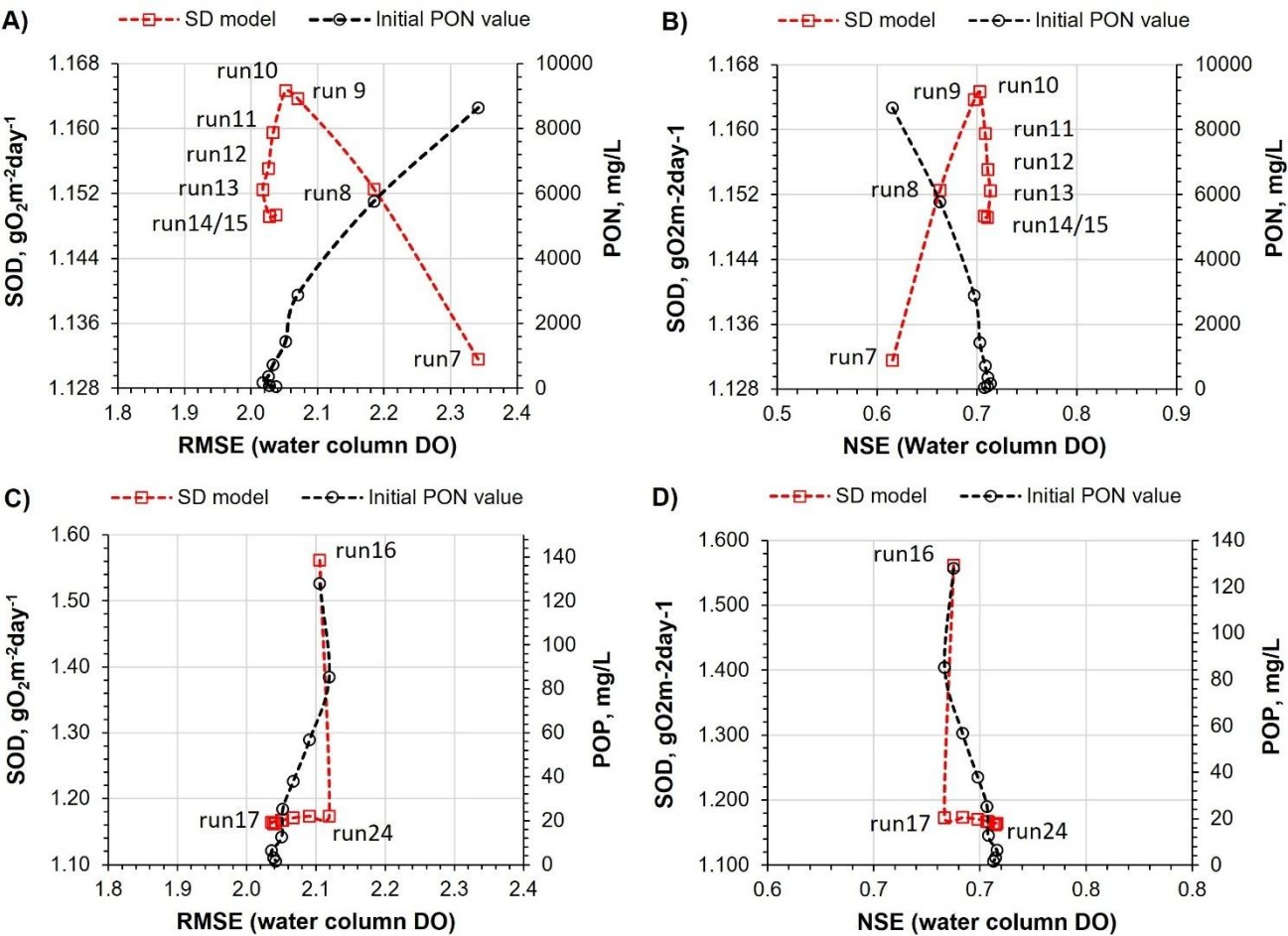

**Figure 10: A) SOD values from the reservoir bottom layer, predicted by the SD model for Runs 7 to 15, compared with the RMSE obtained between the predicted water column DO profiles and the mean initial PON values (across all sites) for each run B) Similar to A) but considering the NSE metric. C) SOD values from the reservoir bottom layer, predicted by the SD model for Runs 16 to 24, compared with the RMSE obtained between the predicted water column DO profiles and the mean initial POP values (across all sites) for each run. D) Similar to C) but considering the NSE metric.**

Figure 11 shows the RMSE (Fig. 11A) and the NSE (Fig. 11B) values between observed and predicted water column DO profiles for all models: SD model (Runs 1 to 6), Zero-order model and Hybrid model, each with six different SOD values ranging from 0.5 to 3.0 g/m²day, along with the corresponding reservoir SOD values. Additionally, this figure illustrates how the First-order model varies with the initial sediment concentration. Among the four models evaluated, the Hybrid model

demonstrated the best overall performance in predicting DO concentrations in the reservoir. With an average SOD of 1.49 g $O_2/m^2day$, the hybrid model achieved the lowest RMSE (1.87 mg/L) and highest NSE (0.76), demonstrating superior predictive accuracy. The Zero-order model followed closely, reaching optimal performance at an average zero-order SOD of 1.43 g $O_2/m^2day$, with an RMSE of 1.965 mg/L and an NSE of 0.732. The SD model also performed well, attaining its best accuracy at an average SOD of 1.07 g $O_2/m^2day$, where the RMSE decreased to 2.011 mg/L and the NSE peaked at 0.716; however, further improvements plateaued beyond this point. In contrast, the First-order model consistently exhibited higher RMSE values (ranging from 2.15 mg/L to 2.22 mg/L) and lower NSE values (between 0.66 and 0.68), regardless of the initial sediment concentration. Moreover, its SOD at the bottom layer remained relatively stable, indicating limited sensitivity to input variations. Overall, these results underscore the hybrid model's robustness and accuracy, followed by the Zero-order and SD models, while the First-order model demonstrated the weakest performance in this context.

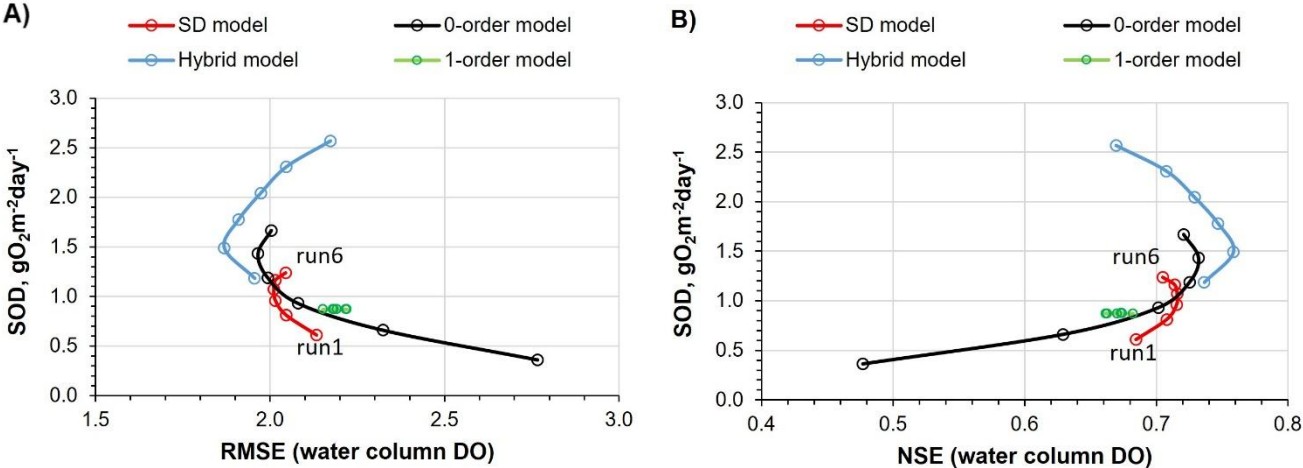

**Figure 11. (A) RMSE between observed and simulated DO profiles in the water column for all models: the SD model (Runs 1–6), the Zero-order model, the Hybrid model with six SOD values ranging from 0.5 to 3.0 g $O_2/m^2day$, and the First-order model with initial sediment organic matter concentrations from 0.0 to 3.0 g $m^{-2}$. (B) Same as (A), but using the Nash–Sutcliffe Efficiency (NSE) as the performance metric.**

**3.4 Inflow Organic Matter and Phosphorus Load Reduction Scenarios**

The results reveal clear differences in model sensitivity to inflow load reductions, with the First-order and Hybrid models exhibiting a stronger response compared to the SD and Zero-order models (Figures 12 and 13). The SD model showed minimal change, indicating limited sensitivity to external loading (Figures 12a and 13a), likely due to strong internal loading feedback from legacy phosphorus and organic matter stored in sediments. The Zero-order model demonstrated limited utility for management scenarios because it is decoupled from the water column, reducing its responsiveness to external changes. The First-order model may overestimate sensitivity as it tends to underestimate internal loading contributions. The Hybrid model, which combines both approaches, is less reactive than the First-order model due to the influence of the Zero-order component, offering a more balanced response. However, the Zero-order SOD component in the Hybrid model depends solely on temperature and remains decoupled from water column conditions; this limitation may gradually reduce the model's accuracy in long-term simulations. These differences in model sensitivity are further reflected in the evolution of average SOD across scenarios (Table 5). While the Zero-order and SD models show virtually no change in bottom-layer SOD under reduced loading conditions, the First-order and Hybrid models register clear declines. The First-order model's SOD drops from 0.87 g $O_2$/m²day to 0.42 g $O_2$/m²day (80% OM reduction) and 0.29 g $O_2$/m²day (80% OM and P reduction) and the Hybrid model from 1.49 g $O_2$/m²day to 1.07 g $O_2$/m².day (80% OM reduction) and 0.94 g $O_2$/m²day (80% OM and P reduction).

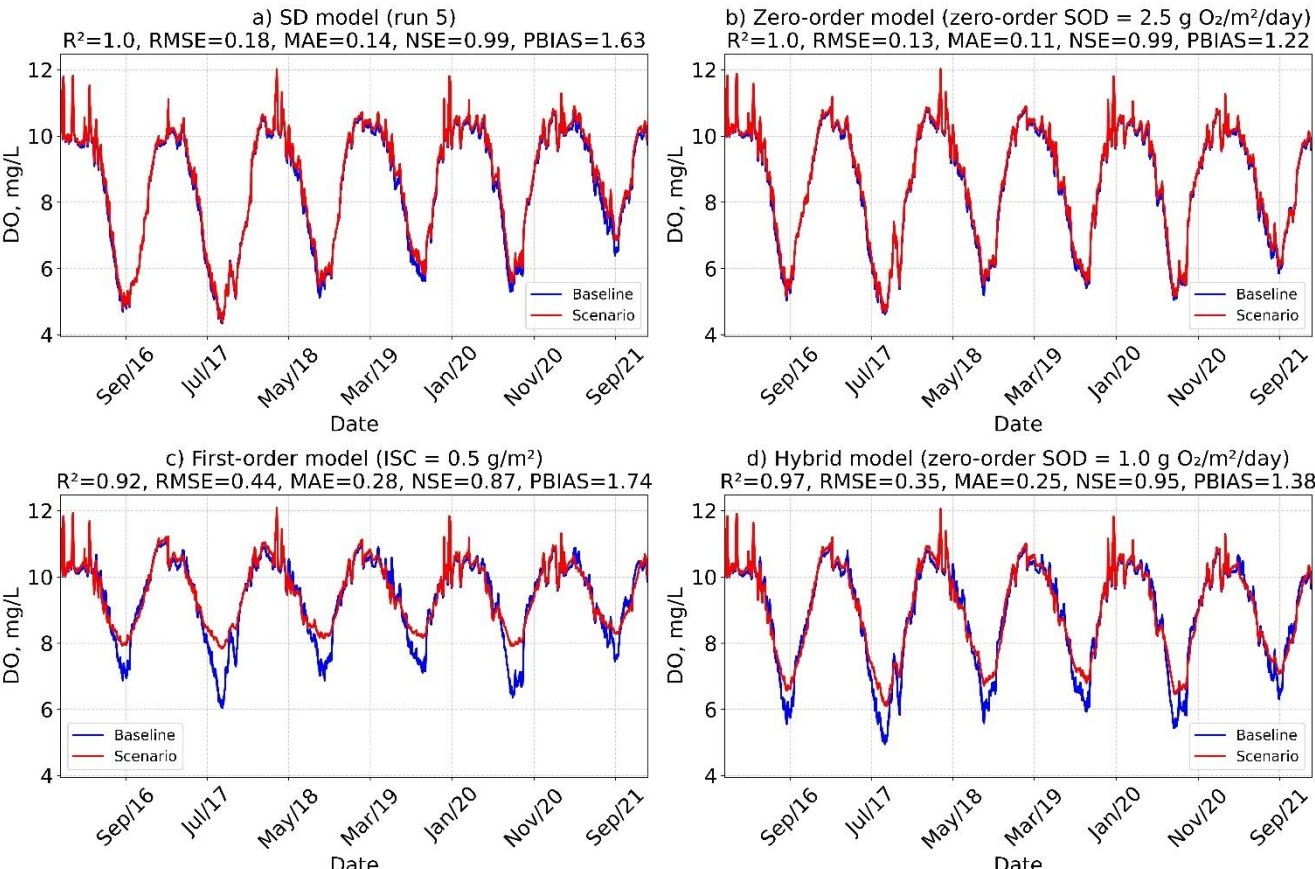

**Figure 12. Time series of DO, averaged across all model layers and segments, for each baseline model scenario: SD model (Run 5), Zero-order model (SOD = 2.5 g O₂/m²day), First-order model (initial sediment concentration = 0.5 g/m²), and Hybrid model (Zero-order SOD = 1.0 g O₂/m²day). The figure compares baseline conditions with an 80% reduction in organic matter inflow load in the main reservoir branch (Branch 1 – Tâmega River). Performance metrics (R², RMSE, MAE, NSE, and PBIAS) are also shown for each case.**

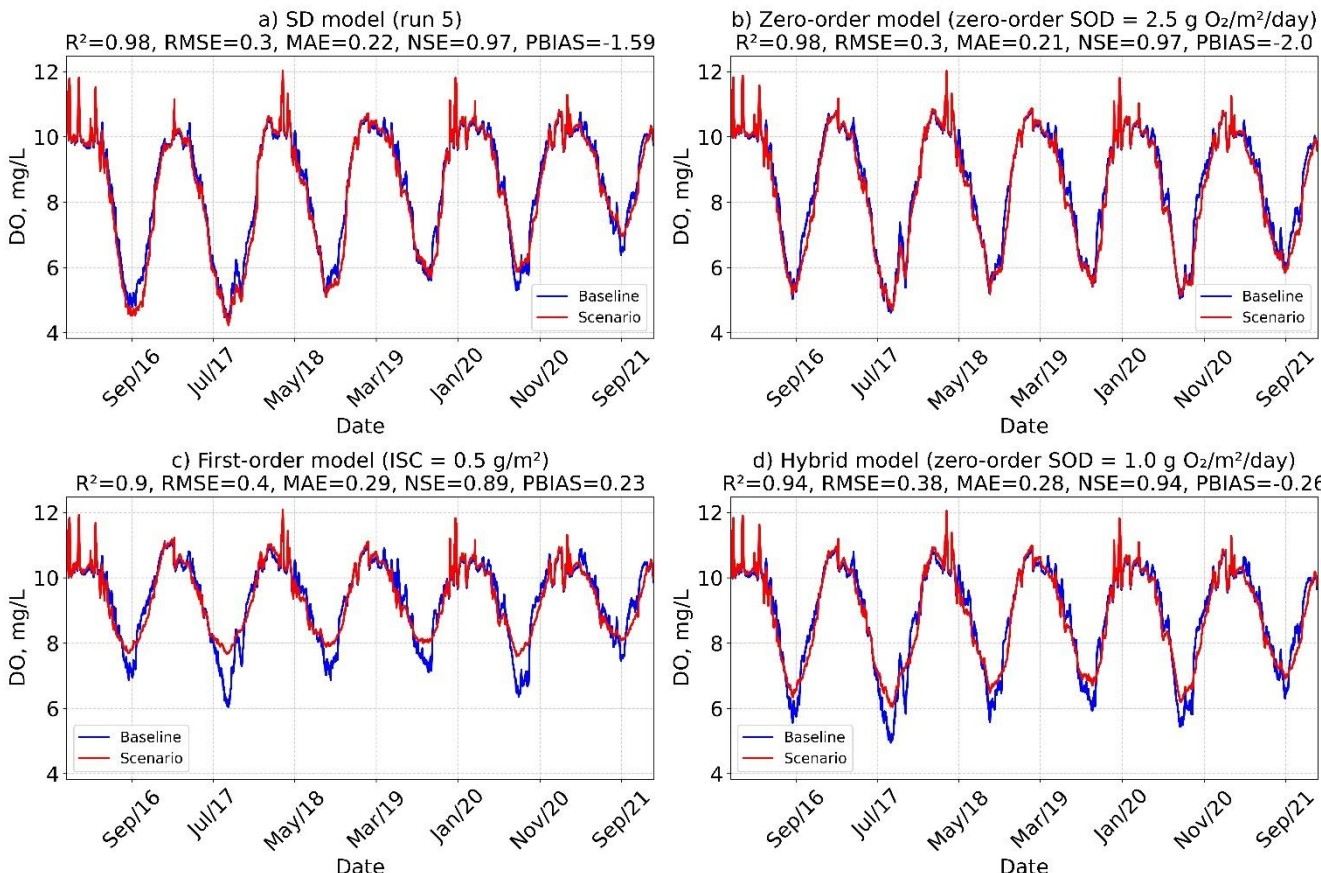

**Figure 13. Time series of DO, averaged across all model layers and segments, for each baseline model scenario: SD model (Run 5), Zero-order model (SOD = 2.5 g O₂/m²day), First-order model (initial sediment concentration = 0.5 g/m²), and Hybrid model (Zero-order SOD = 1.0 g O₂/m²day). The figure compares baseline conditions with an 80% reduction in organic matter and P-PO₄ inflow loads in the main reservoir branch (Branch 1 – Tâmega River). Performance metrics (R², RMSE, MAE, NSE, and PBIAS) are also**
**shown for each case.**

**Table 5. Average sediment oxygen demand (SOD) in the bottom layers of the reservoir, calculated across all segments, for each model under three scenarios: Reference (baseline conditions), 80% reduction in organic matter inflow (OM 80%), and combined 80% reduction in organic matter and phosphorus inflow (OM and P%) in the in the main reservoir branch (Branch 1 – Tâmega River)**

| Scenario | SD model (Run 5) | Zero-order model (SOD = 2.5 g O₂/m²day) | First-order model (ISC = 0.5 g/m²) | Hybrid model (Zero-order SOD = 1.0 g O₂/m.day) | | |
|---|---|---|---|---|---|---|
| | | | | Aggregate | Zero-order term | First-order term |
| **Baseline** | 1.16±0.82 | 1.43±2.12 | 0.87±1.19 | 1.49±2.02 | 0.59± 0.85 | 0.90± 0.75 |
| **OM 80% reduction** | 1.13±0.83 | 1.44±2.12 | 0.42±0.61 | 1.07±1.43 | 0.61± 0.85 | 0.46± 0.75 |
| **OM and P% 80% reduction** | 1.13±0.82 | 1.46±2.11 | 0.29±0.37 | 0.94±1.19 | 0.64± 0.84 | 0.30± 0.42 |

## 4 Discussion

Overall, the temperature and DO predictions for the reservoir boundary conditions (Tâmega river) were quite good: PBIAS: 0.76% and 0.92%, respectively. When a significant number of samples and forcing variables are available the accuracy of machine learning algorithms can be greatly enhanced. This was demonstrated in the studies by Lu et al. (2020), Rajesh and Rehana (2021), and Feigl et al. (2021), where the RMSE for river water temperature prediction reached 1.04ºC, 1.03ºC, and

480 0.58ºC, respectively. The results obtained for alkalinity, conductivity and TSS were also good: Alkalinity-PBIAS: 17.44%; Conductivity - PBIAS: 8.23%; TSS - PBIAS: 11.86%. However, as expected, the PBIAS values obtained for the remaining constituents were not as favorable (Total P- PBIAS: 7.11%; N-NO$_X$- PBIAS: 3.92%; BOD$_5$- PBIAS: 6.93%; Chla- PBIAS: 30%). The modeling of these constituents involves complex biological, chemical, and physical processes that are harder to model accurately. However, except for Chla, the PBIAS values were generally less than 10%, reflecting acceptable levels of

485 bias. Ammonium (N-NH$_4$) was the only parameter for which performance was significantly lower, generating a PBIAS of 28.27%. Moriasi et al. (2015) suggest that $\pm10\leq$PBIAS$\leq \pm 25$ is indicative of a satisfactory model performance.

Based on the RMSE, the overall reservoir calibration results obtained for all constituents with all models for the 2016-2021 period were consistent with the results seen in other studies (see Table A11). The mean RMSE values for Chl-a obtained with all models (SD model (run 5 - baseline): 17.72 µg/L; Zero-order model (zero-order SOD = 2.5 g O₂/m² day - baseline):

17.78 µg/L; First-order model (ISC= 0.5 g/m² - baseline): 14.88 µg/L and the Hybrid model (zero-order SOD=1.0 g O₂/m² day - baseline): 14.88 µg/L) are aligned with the results of other modeling studies (Brito et al., 2018: 62.9 µg/L; Kim et al., 2019: 6.7 to 13.2 µg/L; Tasnim et al., 2021: 0.6 to 27.6 µg/L; Almeida et al., 2023: 19.36 to 25.57 µg/L). For TP, the mean RMSE values were 0.03 mg/L for both the SD model (Run 5 – baseline) and the First-order model (ISC = 0.5 g/m² – baseline), while the Hybrid model (zero-order SOD = 1.0 g O₂/m²day – baseline) showed a slightly higher value of 0.04

495 mg/L. These results fall within the range reported in previous studies, including Brett et al. (2016) at 0.012 mg/L, Kim et al. (2019) between 0.014 and 0.068 mg/L, Tasnim et al. (2021) from 0.005 to 0.036 mg/L, and Almeida et al. (2023) ranging

from 0.07 to 0.09 mg/L. The only exception was the Zero-order model (SOD = 2.5 g $O_2$/m²day – baseline), which overestimated phosphorus export from sediments during the summer months (July to September) of 2018 to 2021, resulting in a notably higher RMSE of 0.1 mg/L. Even with a very low phosphorus release rate from the sediments—representing a fraction of the SOD (0.001)—the Zero-order model still overestimated phosphorus concentrations, particularly during periods of elevated sediment oxygen demand. This suggests that the model may lack the sensitivity needed to accurately simulate low-level sediment-phosphorus interactions under such conditions. The mean RMSE values obtained for TN were lower than the only reference value available in the literature—0.77 mg/L reported by Deliman et al. (2002). Specifically, the SD model (Run 5 – baseline) yielded an RMSE of 0.33 mg/L, the First-order model (ISC = 0.5 g/m² – baseline) produced 0.36 mg/L, and the Hybrid model (zero-order SOD = 1.0 g $O_2$/m²day – baseline) resulted in 0.35 mg/L. The only exception was the Zero-order model (SOD = 2.5 g $O_2$/m²day – baseline), which had a significantly higher RMSE of 0.79 mg/L— slightly exceeding the value reported by Deliman et al., yet still within a comparable range. The RMSE obtained with the SD model (Run 5 - baseline), Zero-order model (zero-order SOD = 2.5 g $O_2$/m²/day - baseline); First-order model (ISC= 0.5 g/m² - baseline) and the Hybrid model (zero order SOD= 1.0 g $O_2$/m²/day - baseline) for DO, 2.01 mg/L, 1.97 mg/L, 2.15 mg/L and 1.87 mg/L respectively) are also in line with the results obtained in other studies (e.g., Deliman et al., 2002: 1.34 mg/L; Brett et al., 2016: 1.2 mg/L; Brito et al., 2018: 7.6 mg/L; Luo et al., 2018: 1.78 mg/L; Tasnim et al., 2021: 2.33 mg/L). In the SD model (Run 5 – baseline), bottom-layer SOD values ranged from 0.015 to 5.152 g $O_2$/m²day ($\mu$ = 1.162; $\sigma$ = 0.823), reflecting moderate variability driven by seasonal biogeochemical processes. In comparison, the Zero-order model (SOD = 2.5 g $O_2$/m²day - baseline) showed a broader but more temperature-driven range, from 0.000 to 15.640 g $O_2$/m²day ($\mu$ = 1.432; $\sigma$ = 2.122). The First-order model (ISC = 0.5 g/m² - baseline) yielded values between 0.000 and 20.000 g $O_2$/m²day, with a much lower mean ($\mu$ = 0.870) and relatively high variability ($\sigma$ = 1.920), consistent with its sensitivity to organic matter loading. The Hybrid model (zero-order SOD = 1.0 g $O_2$/m²day - baseline) incorporated both zero- and first-order processes and produced the widest overall range, from 0.000 to 21.938 g $O_2$/m²day ($\mu$ = 1.491; $\sigma$ = 2.024), highlighting its enhanced responsiveness to both physical (e.g., temperature) and biogeochemical (e.g., organic matter) drivers. The monthly variation in SOD across the four models reveals distinct seasonal patterns influenced by their underlying formulations (Fig. A2). All models show notable peaks in May and October, corresponding to periods of elevated organic matter inflow, while a consistent decline is observed during the summer months (June to August), when external organic inputs are comparatively low. The Zero-order model (baseline SOD = 2.5 g $O_2$/m²·day) exhibits a sharp rise from winter to a peak of 1.919 g $O_2$/m²·day in May, then gradually declines over the summer, before increasing again in October (1.910 g $O_2$/m²day). A similar double-peak pattern is observed in the Hybrid model (zero-order SOD = 1.0 g $O_2$/m²·day, baseline), with SOD reaching 1.715 g $O_2$/m²·day in May and a more pronounced maximum of 2.338 g $O_2$/m²·day in October, reflecting the combined effects of temperature and organic matter availability. The SD model (Run 5 – baseline) shows more moderate seasonal variation, with values dipping to 0.679 g $O_2$/m²·day in August, then rising to 1.501 g $O_2$/m²·day in November, consistent with internal sediment dynamics. The First-order model (ISC = 0.5 g/m², baseline), which is most sensitive to organic matter loading, also mirrors this seasonal structure, peaking in October (1.235 g $O_2$/m²day) after a gradual summer

decline. Collectively, these patterns underscore the importance of organic matter availability—particularly in spring and autumn—as a key driver of SOD across the different modeling approaches. This pattern indicates the model's responsiveness to both organic matter inputs and temperature, leading to a more nuanced representation of seasonal variation compared to the other models. These values are consistent with the SOD values obtained in other studies, such as those of Schnoor and Fruh (1979), which concluded that the SOD values of Lake Lydon B. Johnson (located in the U.S.) ranged from 1.7 to 5.8 g $O_2$/m²day, and of Beutel (2015), which measured SOD values in different locations around Lake Hodges (located in the U.S.) ranging from 0.6 to 2.3 g $O_2$/m²day. It would be useful to be able to compare these results with SOD values measured at different sites within the Torrão reservoir.

It is important to emphasize that this study was primarily designed to evaluate the performance of the sediment diagenesis model. However, by incorporating alternative SOD modeling approaches, it inevitably allowed for a comparative ranking of model performance, highlighting the relative strengths and limitations of each formulation. The performance limitations of the Zero-order and First-order models can be attributed to their structural simplifications. Specifically, the Zero-order model's strong temperature dependence, coupled with its disregard for the dynamics of organic matter loading, reduces its ability to capture temporal variability driven by external inputs. Similarly, the lower accuracy of the First-order model likely stems from its exclusion of anaerobic decay processes and limited representation of sediment biogeochemistry, which becomes especially relevant under low-oxygen conditions. The Hybrid model outperformed all other approaches. Considering the principle of parsimony (Occam's razor) (Burnham and Henderson, 2002), the simpler Hybrid model proved more effective than the complex SD model, making it the preferred choice for simulating SOD dynamics in the reservoir. These findings underscore the importance of selecting models that align with the specific characteristics of the system being studied. Simpler models, such as the Hybrid model, may be adequate for steady-state conditions, short- to medium-term forecasts, or scenarios with limited data. The zero-order SOD component of the Hybrid model relies solely on temperature and is decoupled from the water column; therefore, in long-term simulations, this limitation can gradually undermine the model's accuracy. In contrast, the SD model may be more appropriate when the goal is to explore system-wide feedbacks and temporal dynamics over extended periods—especially those involving sediment accumulation and nutrient cycling— where it may provide valuable insight into underlying processes, provided that sufficient observational data become available to support its additional state variables. Moreover, a model's effectiveness heavily depends on the user's familiarity with its structure and their skill in calibration. Yet, it is unrealistic to expect researchers to master the implementation of every available modeling approach. As such, comparisons between models should be interpreted carefully, acknowledging the influence of user expertise on performance outcomes (Piccolroaz et al. 2024). Overall, to strengthen the analysis, it is recommended that users apply all available SOD modeling approaches in the case of the CE-QUAL-W2 model and assess the model's behavior. This comprehensive evaluation provides a solid foundation for further modeling efforts and helps ensure that the chosen approach is well-suited to the system's specific conditions and objectives.

The results also revealed that the particulate fraction of organic carbon in the reservoir sediments corresponded to 80% of the TOC. This value is small compared to the results obtained for Taihu Lake by Yu et al. (2022), where the ratio of POP to

TOC varied from 97.85% to 89.53%. However, this value (80%) was obtained indirectly through the analysis of the reservoir's predicted SOD values as a function of different initial POC values and may, therefore, reflect other sources of uncertainty, such as inflow organic matter characterization. Given the fact that the magnitude of TOC in the sediment can be affected by numerous factors, including water column productivity, terrestrial inputs of organic materials, sediment properties, and microbial activity rates (Gireeshkumar et al., 2013), and that, partly due to differences in reservoir productivity and morphology, the spatial distribution and sources of organic carbon vary greatly across regions (Anderson et al., 2009), it is reasonable to assume that the only way to accurately assess the POC prediction is by monitoring the reservoir POC content. Furthermore, this study has highlighted the need to expand research to additional waterbodies across diverse regions to improve our understanding of the CE-QUAL-W2 diagenesis model's performance under varying environmental conditions. This includes evaluating its applicability in long-term scenarios, which are essential for capturing cumulative sediment dynamics and climate-driven trends. Additional SOD monitoring studies need to be conducted in lakes and reservoirs and extended to other latitudes, with particular focus on the chemical characterization of sediments and the definition of sediment burial rates.

## 5 Conclusions

This research evaluates the performance of the CE-QUAL-W2 v4.5 sediment diagenesis model in simulating water temperature, dissolved oxygen, total phosphorus, total nitrogen, chlorophyll-a, and biochemical oxygen demand in a Portuguese reservoir over the period from 2016 to 2021. Calibration was based on 35 sets of observed temperature and dissolved oxygen profiles, supplemented by six annual measurements of total nitrogen, total phosphorus, chlorophyll-a, and biochemical oxygen demand collected at various depths. To evaluate model accuracy, three alternative sediment oxygen demand formulations — a Zero-order model, a First-order model, and a Hybrid approach combining features of both — were also applied and compared. The Hybrid model consistently outperformed the other formulations, striking an effective balance between accuracy and simplicity. It therefore represents the most suitable choice for modeling the reservoir. In contrast, the Zero- and First-order models exhibited limitations related to temperature dependence and inadequate sediment process representation, respectively. Simpler models, such as the Hybrid model, may be adequate for steady-state conditions, short- to medium-term forecasts, or scenarios with limited data. In contrast, the SD model — despite its good performance — may be more appropriate when the goal is to explore system-wide feedbacks and temporal dynamics over extended periods, especially in cases involving sediment accumulation and nutrient cycling. In such contexts, it may offer valuable insights, provided that sufficient observational data are available to support its additional state variables. Overall, the study reinforces the importance of choosing models based on site characteristics, available data, and simulation goals. Future work should broaden the evaluation of these models across various waterbodies and extended timeframes, while highlighting the need for enhanced sediment monitoring to support detailed process-based modelling.

# Appendix A

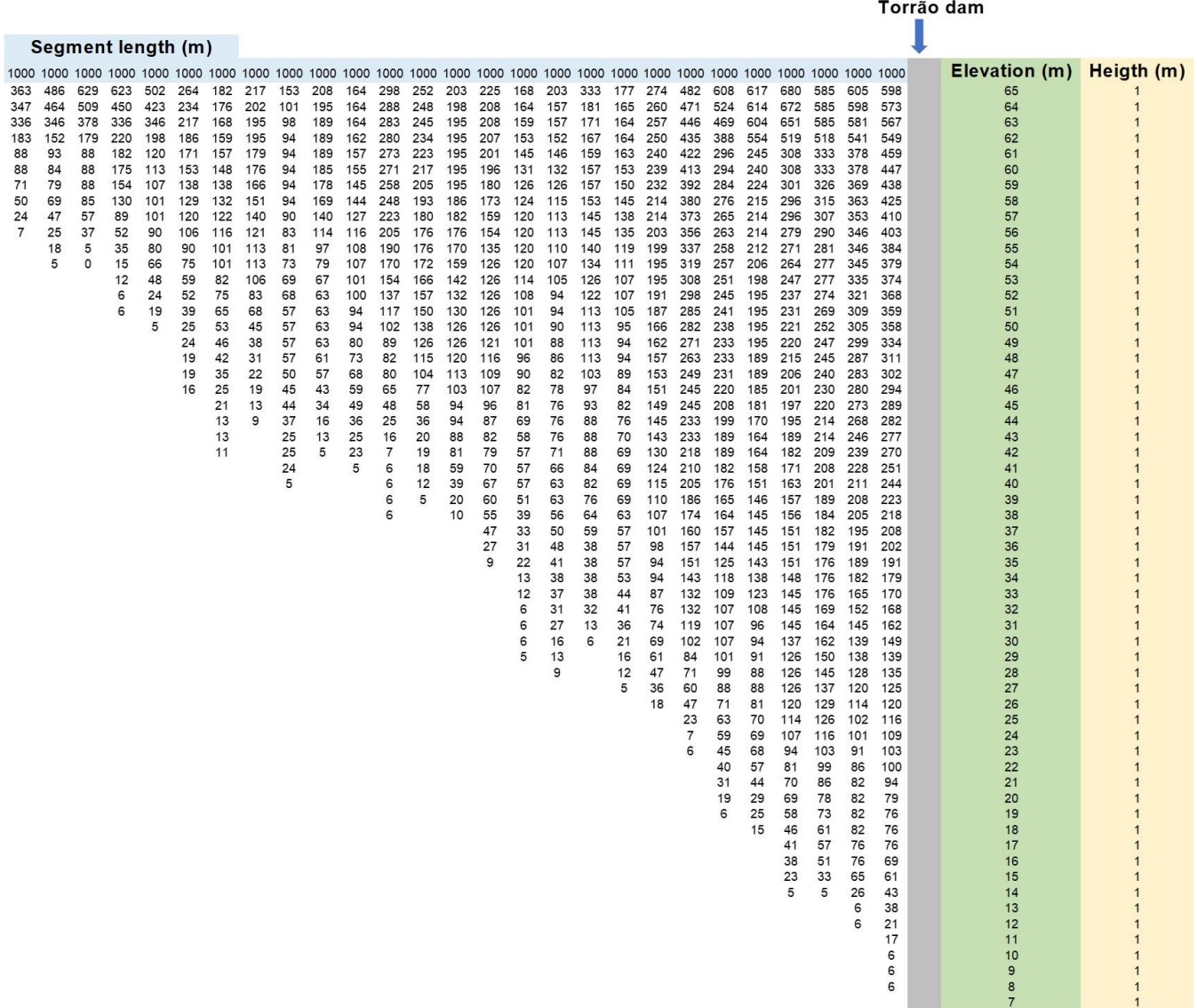

**Figure A1: CE-QUAL-W2 bathymetry - Cross section of the Tâmega River with the average segment width**

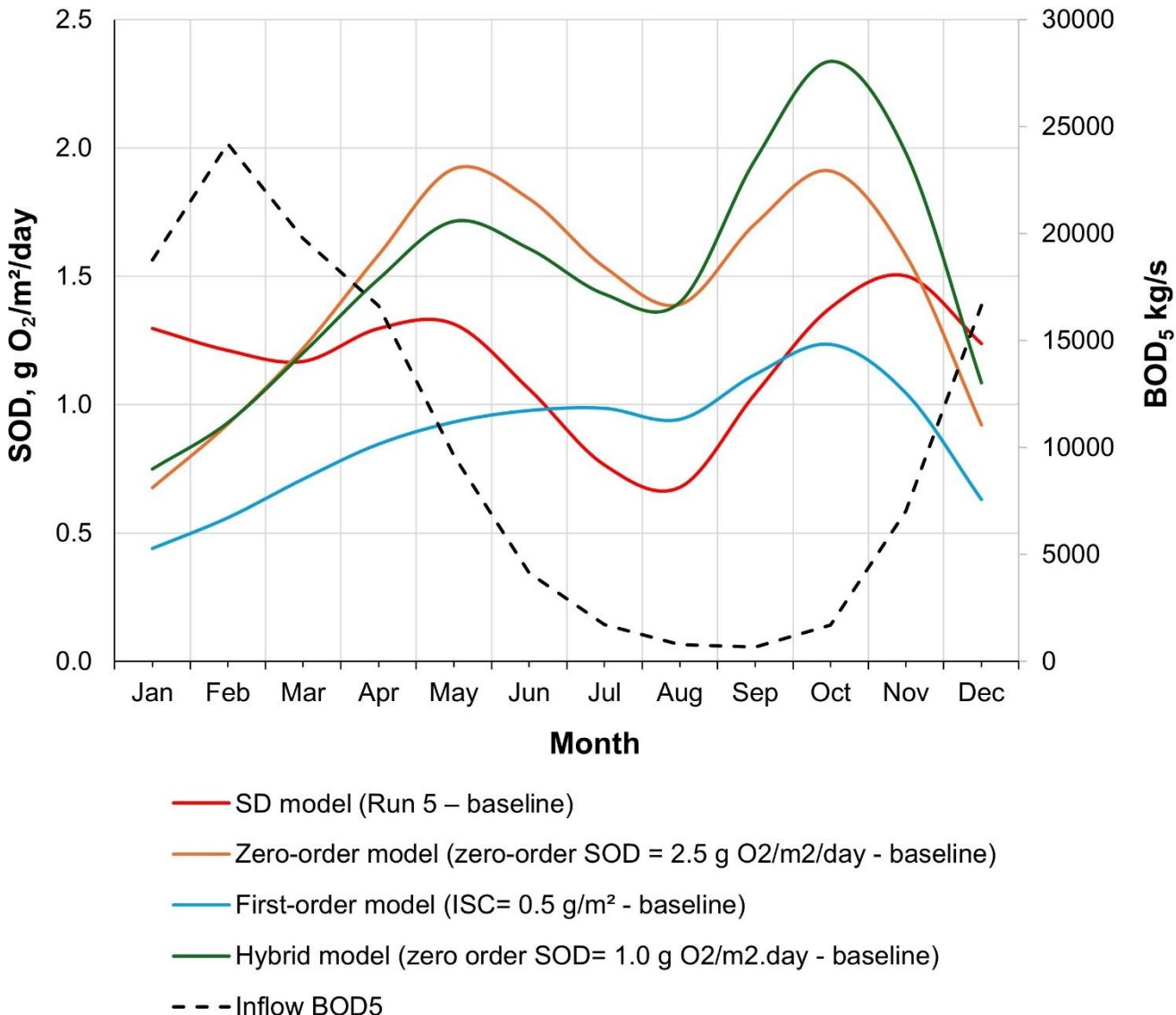

**Figure A2: Average monthly sediment oxygen demand (SOD) at the reservoir bottom layer predicted by the SD model (Run 5 - baseline), Zero-order model (baseline: SOD = 2.5 g $O_2$/m²day), First-order model (baseline: ISC = 0.5 g/m²), and Hybrid model (baseline: zero-order SOD = 1.0 g $O_2$/m²day). Also shown is the inflow $BOD_5$ load from the reservoir's main branch.**

**Table A1. Model metrics. Characterization of Tâmega river inflow**

| Constituent | Model | $R^2$ | PBIAS | RMSE | MAE | Input features | Input sample number | Sample Time period |
|---|---|---|---|---|---|---|---|---|
| Water temperature | LOADEST | 0.81 | 5.14 | 2.61 | 1.86 | Flow; AirT; WT | 21 | 2016-2021 |
| | XGBOOST | 0.85 | 0.76 | 2.19 | 1.76 | Air T; WT | | |
| | SVR | 0.87 | 3.77 | 2.07 | 1.60 | Air T; WT | | |
| DO | LOADEST | 0.11 | 5.02 | 1.32 | 1.03 | Flow, DO | 78 | 2010-2021 |
| | XGBOOST | 0.83 | 1.50 | 0.56 | 0.33 | WT; DO; Month | | |
| | SVR | 0.91 | 0.92 | 0.40 | 0.26 | WT; DO; Month | | |
| Total P | LOADEST | 0.21 | 7.11 | 0.02 | 0.01 | Flow, Total P | 47 | 2010-2021 |
| | XGBOOST | 0.12 | - | 45.72 | 32.73 | WT; Total P; Month | | |
| | SVR | - | - | 38.74 | 38.74 | WT; Total P; Month | | |
| N-NH$_4$ | LOADEST | 0.11 | 43.64 | 0.02 | 0.02 | Flow; N-NH$_4$ | 46 | 2010-2021 |
| | XGBOOST | 0.26 | 32.20 | 0.02 | 0.02 | WT; N-NH$_4$; Month | | |
| | SVR | 0.03 | 28.27 | 0.02 | 0.02 | WT; N-NH$_4$; Month | | |
| N-NO$_X$ | LOADEST | 0.11 | 10.88 | 0.18 | 0.14 | Flow; N-NO$_X$ | 63 | 2010-2021 |
| | XGBOOST | 0.15 | 3.92 | 0.16 | 0.12 | WT; N-NO$_X$; Month | | |
| | SVR | 0.26 | 7.48 | 0.16 | 0.13 | WT; N-NO$_X$; Month | | |
| BOD$_5$ | LOADEST | 0.19 | 6.93 | 0.98 | 0.83 | Flow; BOD$_5$ | 77 | 2010-2021 |
| | XGBOOST | - | 782.16 | 16.31 | 16.28 | WT; BOD$_5$; Month | | |
| | SVR | 0.14 | 1438.34 | 30.64 | 29.93 | WT; BOD$_5$; Month | | |
| Chlorophyll-a | LOADEST | 0.65 | 30.00 | 6.27 | 4.96 | Flow; Chl-a | 49 | 2010-2021 |
| | XGBOOST | 0.12 | 774.02 | 39.12 | 29.36 | WT; Chl-a; Month | | |
| | SVR | 0.02 | 345.72 | 15.05 | 13.25 | WT; Chl-a; Month | | |
| Alkalinity | LOADEST | 0.26 | 17.44 | 6.59 | 6.30 | WT; Chl-a; Month | 31 | 2013-2021 |
| | XGBOOST | 0.26 | 471.67 | 88.57 | 57.64 | WT; Alkalinity; | | |
| | SVR | 0.20 | 242.35 | 37.97 | 30.52 | WT; Alkalinity; | | |
| Conductivity | LOADEST | 0.80 | 8.23 | 8.81 | 7.69 | Flow; Conductivity | 77 | 2010-2021 |
| | XGBOOST | 0.32 | 184.79 | 133.16 | 120.16 | WT; Conductivity; | | |
| | SVR | 0.02 | 17.67 | 21.96 | 17.32 | WT; Conductivity; | | |
| TSS | LOADEST | 0.00 | 11.86 | 1.83 | 1.45 | Flow; SST | 78 | 2010-2021 |
| | XGBOOST | 0.19 | 7.06 | 1.96 | 1.38 | WT; SST; Month | | |
| | SVR | 0.24 | 8.44 | 1.62 | 1.24 | WT; SST; Month | | |

**Table A2. Rates and constants: Hydraulic coefficients**

| Rates and constants | Value |
|---|---|
| Transport solution scheme | Ultimate |
| Time-weighting for vertical advection scheme | 0.55 |
| Longitudinal eddy viscosity (m²/s) | 1.0 |
| Longitudinal eddy diffusivity (m²/s) | 1.0 |
| Coefficient of bottom heat exchange (W/m²/s) | 0.3 |
| Sediment temperature (°C) | 14.0 |
| Interfacial friction factor | 0.0 |
| Heat lost to sediments that is added back to water column | 0.0 |
| Vertical eddy viscosity | W2 |
| Maximum value of vertical eddy speed (m²/s) | 1.0 |
| Bottom friction solution | CHEZY |
| Wind roughnessheight (m) | 0.001 |

**Table A3. Rates and constants: Extinction coefficients**

| Rates and constants | Value |
|---|---|
| Extinction for pure water, m$^{-1}$ | 0.25 |
| Suspended solids light extinction, 1/m | 0.01 |
| EXOM - extinction organic matter, 1/(m mg/l) | 0.2 |
| Fraction of incident solar radiation absorbed at the water surface (long-wave components of short-wave solar) | 0.45 |

**Table A4. Rates and constants: phytoplankton (Diatoms)**

| Rates and constants | Value |
|---|---|
| Maximum algal growth rate (day$^{-1}$) | 3 |
| Maximum algal respiration rate (day$^{-1}$) | 0.04 |
| Maximum algal excretion rate (day$^{-1}$) | 0.04 |
| Maximum algal mortality rate (day$^{-1}$) | 0.1 |
| Algal settling rate (day$^{-1}$) | 0.1 |
| Algal half-saturation for phosphorus-limited growth, g m$^{-3}$ | 0.003 |
| Algal half-saturation for nitrogen limited growth, g m$^{-3}$ | 0.014 |
| Fraction of biomass going to POM at death | 0.8 |
| AT1 lower temperature for algal growth, °C | 10 |
| AT2 lower temperature for maximum algal growth, °C | 30 |
| AT3 upper temperature for maximum algal growth, °C | 35 |
| AT4 upper temperature for algal growth, °C | 40 |
| Fraction of growth rate at AT1 temperature | 0.1 |
| Fraction of maximum growth rate at AT2 | 0.99 |
| Fraction of maximum growth rate at AT3 | 0.99 |
| Fraction of growth rate at AT4 | 0.1 |
| Chlorophyll a to algae biomass ratio | 0.065 |

**Table A5. Rates and constants: organic matter**

| Rates and constants | Value |
|---|---|
| Labile DOM decay rate, day$^{-1}$ | 0.3 |
| Refractory DOM decay rate, day$^{-1}$ | 0.001 |
| Labile to refractory DOM decay rate, day$^{-1}$ | 0.01 |
| Labile POM decay rate, day$^{-1}$ | 0.08 |
| Refractory POM decay rate, day$^{-1}$ | 0.01 |
| Labile to refractory POM decay rate, day$^{-1}$ | 0.001 |
| POM settling rate, m day$^{-1}$ | 0.5 |
| Lower temperature for organic matter decay, °C, OMT1 | 4 |
| Upper temperature for organic matter decay, °C, OMT2 | 25 |
| Fraction of organic matter decay rate at OMT1 | 0.1 |
| Fraction of organic matter decay rate at OMT2 | 0.99 |

**Table A6. Rates and constants: nutrients**

| Rates and constants | Value |
|---|---|
| PO4R Sediment release rate of phosphorus, fraction of SOD | 0.015 |
| PARTP Phosphorus partitioning coefficient for suspended solids | 0.0 |
| NH4R Sediment release rate of ammonium, fraction of SOD | 0.15 |
| NH4DK Ammonium decay rate, day$^{-1}$ | 0.05 |
| NH4T1 Lower temperature for ammonia decay, ºC | 5 |
| NH4T2 Lower temperature for maximum ammonia decay, ºC | 25 |
| NH4K1 Fraction of nitrification rate at NH4T1 | 0.1 |
| NH4K2 Fraction of nitrification rate at NH4T2 | 0.99 |
| O2NH4 Oxygen stoichiometry for nitrification (mg $O_2$/mg N) | 4.57 |
| NO3DK Water column denitrification rate or nitrate decay rate, day$_{-1}$ | 0.05 |
| NO3S Nitrate loss velocity to the sediments because of sediment denitrification, m day$_{-1}$ | 0 |
| FNO3SED Fraction of NO3-N diffused into the sediments that becomes part of organic N in the sediments (The rest is denitrified.) | 0.37 |
| NO3T1 Lower temperature for nitrate decay, ºC | 5 |
| NO3T2 Lower temperature for maximum nitrate decay, ºC | 25 |
| NO3K1 Fraction of denitrification rate at NO3T1 | 0.1 |
| NO3K2 Fraction of denitrification rate at NO3T2 | 0.99 |

**Table A7. Rates and constants: SOD rates**

| Rates and constants | Value |
|---|---|
| SEDCI Initial first order sediment concentration, g m$^{-2}$ | 0.5 |
| SEDK First order sediment decay rate, day$^{-1}$ | 0.1 |
| SEDS First order sediment settling or focusing rate, m day$^{-1}$ | 0.1 |
| SEDBR First order sediment burial rate, day$^{-1}$ | 0.01 |
| DYNSEDK Turns ON/OFF dynamic calculation of the first order sediment model decay rate | ON |
| SODT1 Lower temperature for zero-order SOD or first-order sediment decay, ºC | 4 |
| SODT2 Upper temperature for zero-order SOD or first-order sediment decay, ºC | 30 |
| SODK1 Fraction of SOD or sediment decay rate at lower temperature | 0.1 |
| SODK2 Fraction of SOD or sediment decay rate at upper temperature | 0.99 |

**Table A8. Rates and constants: Sediment diagenesis model (considering 5 regions)**

"Initial sediment temperature ºC",10,10,10,10,10,
"Initial sediment pH (only used if include modeling dynamic pH)",7,7,7,7,7
"Initial POC (total) concentration
mgC/L",4301,15437,17126,16051,19200,12902
"Initial PON (total) concentration mgN/L",1000,1000,1000,1000,1000
"Initial POP (total) concentration mgP/L",0.001,0.001,0.001,0.001,0.001
"Initial SO$_4$ concentration mgS/L",0.1,0.1,0.1,0.1,0.1
"Initial dissolved NH$_4$ concentration mgN/L",0.01,0.01,0.01,0.01,0.01
"Initial dissolved NO$_3$ concentration mgN/L",0.0,0.0,0.0,0.0,0.0
"Initial total PO$_4$ concentration mgP/L",0.00,0.00,0.00,0.00,0.00
"Initial dissolved H2S concentration mgS/L",0.1,0.1,0.1,0.1,0.1
"Initial CH$_4$ concentration mgC/L",0.092,0.092,0.092,0.092,0.092
"Initial TIC concentration mgC/L",4.011,4.011,4.011,4.011,4.011
"Initial Alkalinity concentration for each region mg/l as CaCO3",50,50,50,50,50
"Initial Ferrous Iron concentration for each region mgFe/l",0.1,0.1,0.1,0.1,0.1
"Initial Iron Oxyhydroxide concentration for each region
mgFe/l,",0.1,0.1,0.1,0.1,0.1
"Initial Mn(II) concentration for each region mgMn/l,",0.1,0.1,0.1,0.1,0.1
"Initial manganese dioxide concentration for each region
mgMn/l,",0.1,0.1,0.1,0.1,0.1
"Number of regions for different diagenesis related rates",1
,Region1,Region2,Region3,Region4,Region5
"Starting segment for regions",2
"Ending segment for regions",21
"Fraction of labile POC",0.1
"Fraction of refractory POC",0.89
"Fraction of labile PON",0.1
"Fraction of refractory PON",0.89
"Fraction of labile POP",0.1
"Fraction of refractory POP",0.89
"Pore water diffusion coefficient m$^2$/d",0.0025
"Particle mixing velocity between aerobic and anaerobic layers m$^2$/d",0.00006
"Burial velocity m/d",0.00000685
"CH$_4$ production calculation method (0: Analytical  1: Numerical)",1,1
"DO threshold for aerobic layer oxidation rates mgO$_2$/L",2,2
"Nitrification rate in aerobic layer at DO below threshold m/d",0.3
"Nitrification rate in aerobic layer at DO above threshold m/d",0.3
"Denitrification rate in aerobic layer at DO below threshold m/d",0.1
"Denitrification rate in aerobic layer at DO above threshold m/d",0.1
"Denitrification rate in anerobic layer m/d",0.1,0.1
"CH$_4$ oxidation rate in aerobic layer m/d",0.7,0.7
"Half-saturation oxygen constant for CH$_4$ oxidation mg-O$_2$/L",0,0
"Nitrification half-saturation constant for NH$_4$ in aerobic layer mgN/L",0.728
"Nitrification half-saturation constant for O$_2$ in aerobic layer mgO$_2$/L",0.37
"Temperature coefficient for pore water diffusion between layers",1.08
"Temperature coefficient for particle mixing diffusion coefficient",1
"Temperature coefficient for nitrification",1.123
"Temperature coefficient for denitrification",1.08
"Temperature coefficient for methane oxidation",1.079
"SO4 concentration above which sulfide over methane is produced mgS/L",20
"H2S oxidation rate in aerobic layer m/d",0.2
"Temperature coefficient for H$_2$S oxidation",1.079
"H2S oxidation normalization constant for O2 mgO$_2$/L",4
"Diagenesis rate for labile POC (G1) 1/d",0.035
"Diagenesis rate for refractory POC (G2) 1/d",0.0018
"Diagenesis rate for inert/slow refractory POC (G3) 1/d",0.0001
"Diagenesis rate for labile PON (G1) 1/d",0.035
"Diagenesis rate for refractory PON (G2) 1/d",0.0018
"Diagenesis rate for inert/slow refractory PON (G3) 1/d",0.0001
"Diagenesis rate for labile POP (G1) 1/d",0.035
"Diagenesis rate for refractory POP (G2) 1/d",0.0018

```
"Diagenesis rate for inert/slow refractory POP (G3) 1/d",0.0001
"Temperature coefficient for labile POC",1.1
"Temperature coefficient for refractory POC",1.15
"Temperature coefficient for inert/slow refractory POC",1
"Temperature coefficient for labile PON",1.1
"Temperature coefficient for refractory PON",1.15
"Temperature coefficient for inert/slow refractory PON",1,1
"Temperature coefficient for labile POP",1.1,1.1
"Temperature coefficient for refractory POP",1.15,1.15
"Temperature coefficient for inert/slow refractory POP",1
"PO4 sorption coefficient in anaerobic layer m3/kg",0.02
"Incremental PO4 partition coefficient",0,0
"Critical oxygen concentration for incremental sorption mg-O2/L",0.01
"NH4 sorption coefficient in aerobic layer m3/kg",0.001
"NH4 sorption coefficient in anaerobic layer m3/kg",0.001
"H2S sorption coefficient in aerobic layer m3/kg",0.1
"H2S sorption coefficient in anaerobic layer m3/kg",0.1
"Algorithm for POM resuspension (0: Wind induced resuspension, 1: Bottom
scour resuspension) Only used if Include POM resuspension is TRUE.",1,0
"Fe(II) sorption coefficient in aerobic layer m3/g",0.00005
"Fe(II) sorption coefficient in anaerobic layer m3 /g",0.01
"Mn(II) sorption coefficient in aerobic layer m3 /g",0.00005
"Mn(II) sorption coefficient in anaerobic layer  m3 /g",0.01
"Write sediment fluxes", TRUE.
"Frequency of output days",1
```

**Table A9. Metrics between observed and predicted values for both models. The predicted values were compared with observed values at three different depths: (a) an integrated sample between the reservoir surface and an average depth of 5.8 meters, (b) an average depth of 23 meters, and (c) an average depth of 43.7 meters. The values shown in this table represent the mean value of the metrics obtained for each date and the corresponding standard deviation.**

| Constituent | SD model (run 5 - baseline) | | | |
| --- | --- | --- | --- | --- |
| | $R^2$ | PBIAS | RMSE | MAE |
| Water temperature | 0.95 | -3.34 | 1.31 | 1.05 |
| DO | 0.79 | -9.68 | 2.01 | 1.40 |
| Total N | 0.19 | -15.85 | 0.33 | 0.27 |
| Total P | 0.09 | -11.44 | 0.03 | 0.01 |
| BOD$_5$ | 0.02 | -50.92 | 3.24 | 1.58 |
| Chl-a | 0.14 | 51.62 | 17.12 | 11.51 |
| Constituent | Zero-order model (zero order SOD = 2.5 g O$_2$/m$^2$/day - baseline) | | | |
| | $R^2$ | PBIAS | RMSE | MAE |
| Water temperature | 0.95 | -3.34 | 1.32 | 1.06 |
| DO | 0.84 | -5.91 | 1.69 | 1.12 |
| Total N | 0.22 | -1.07 | 0.79 | 0.46 |
| Total P | 0.26 | 103.43 | 0.10 | 0.04 |
| BOD$_5$ | 0.03 | -50.93 | 3.24 | 1.58 |
| Chl-a | 0.09 | 48.89 | 17.78 | 12.17 |
| Constituent | First-order model (ISC= 0.5 g/m$^2$ - baseline) | | | |
| | $R^2$ | PBIAS | RMSE | MAE |
| Water temperature | 0.95 | -3.18 | 1.31 | 1.05 |
| DO | 0.83 | 9.21 | 1.84 | 1.28 |
| Total N | 0.14 | -24.73 | 0.36 | 0.28 |
| Total P | 0.10 | -3.71 | 0.03 | 0.02 |
| BOD$_5$ | 0.00 | -53.34 | 3.24 | 1.59 |
| Chl-a | 0.06 | -60.39 | 14.88 | 8.49 |
| Constituent | Hybrid model (zero order SOD= 1.0 g O$_2$/m$^2$/day - baseline) | | | |
| | $R^2$ | PBIAS | RMSE | MAE |
| Water temperature | 0.95 | -3.20 | 1.31 | 1.05 |
| DO | 0.87 | -2.34 | 1.52 | 1.03 |
| Total N | 0.31 | -18.75 | 0.35 | 0.28 |
| Total P | 0.27 | 36.49 | 0.04 | 0.02 |
| BOD$_5$ | 0.01 | -51.93 | 3.25 | 1.61 |
| Chl-a | 0.06 | -59.55 | 14.95 | 8.52 |

**Table A10. Metrics between observed and predicted values for SD model (run 4). Water temperature and DO metrics were obtained from 36 observed and predicted profiles. The predicted values for the remaining constituents were compared with observed values at three different depths: (a) an integrated sample between the reservoir surface and an average depth of 5.8 meters, (b) an average depth of 23 meters and (c) an average depth of 43.7 meters. The values in this table represent the mean value of the metrics obtained at each date and the corresponding**
**standard deviation or, in the case of water temperature and DO, the mean value of the metrics obtained for each profile and the standard deviation.**

| Constituent | SD model (Run 4) | | | |
|---|---|---|---|---|
| | $R^2$ | PBIAS | RMSE | MAE |
| Water temperature | 0.95 | -3.45 | 1.34 | 1.07 |
| DO | 0.79 | -8.20 | 1.98 | 1.37 |
| Total N | 0.18 | -15.52 | 0.33 | 0.27 |
| Total P | 0.09 | -11.18 | 0.03 | 0.01 |
| BOD5 | 0.02 | -50.93 | 3.24 | 1.58 |
| Chl-a | 0.15 | 52.95 | 16.97 | 11.48 |

**Table A11. Root mean square error values obtained with different models and across different time frames**

| Constituent | Model | Simulation | RMSE | Author |
|---|---|---|---|---|
| Water temperature | CE-QUAL-W2 | 1984-1991 | 2.95 ºC | Deliman et al., 2002 |
| | CE-QUAL-W2 | 2005-2010 | 1.93 ºC | Kim and Kim, 2006 |
| | CE-QUAL-W2 | 1991-2000 | 0.56 ºC | Zhang et al., 2015 |
| | CE-QUAL-W2 | 2011-2013 | < 2.0 ºC | Lindenschmidt et al., 2019 |
| | Delft3D-FLOW | 2015-2017 | 0.96 to 1.0ºC | Piccioni, et al. 2020 |
| | CE-QUAL-W2 | 2001-2011 | 1.80 ºC | Brito et al., 2018 |
| | CE-QUAL-W2 | 2010 | 2.36 ºC | Liu et al., 2019 |
| | MINLAKE2020 | 2007-2009 | 1.51 ºC | Tasnim et al., 2021 |
| | CE-QUAL-W2 | 2000-2019 | 3.01-3.17 ºC | Almeida et al., 2023 |
| DO | CE-QUAL-W2 | 1984-1991 | 1.34 mg/l | Deliman et al., 2002 |
| | CE-QUAL-W2 | 1991-2000 | 0.61 mg/l | Zhang et al., 2015 |
| | CE-QUAL-W2 | 2001 | 1.2 mg/l | Brett et al., 2016 |
| | CE-QUAL-W2 | 2001-2011 | 7.6 mg/l | Brito et al., 2018 |
| | DYRESM 4.0 | 2009-2011 | 1.78 mg/l | Luo et al., 2018 |
| | MINLAKE2020 | 2007-2009 | 2.33 mg/l | Tasnim et al., 2021 |
| | CE-QUAL-W2 | 2000-2019 | 2.22-3.46 mg/l | Almeida et al., 2023 |
| Total P | CE-QUAL-W2 | 2001 | 0.012 mg/l | Brett et al., 2016 |
| | CE-QUAL-W2 | 2005-2010 | 0.014 to 0.068 mg/l | Kim et al., 2019 |
| | MINLAKE2020 | 2007-2009 | 0.005 to 0.036 mg/l | Tasnim et al., 2021 |
| | CE-QUAL-W2 | 2000-2019 | 0.07-0.09 mg/l | Almeida et al., 2023 |
| Total N | CE-QUAL-W2 | 1984-1991 | 0.77 mg/L | Deliman et al., 2002 |
| $BOD_5$ | CE-QUAL-W2 | 2000-2019 | 3.06-4.83 mg/l | Almeida et al., 2023 |
| Chl-a | CE-QUAL-W2 | 1991-2000 | 1.08 µg/l | Zhang et al., 2015 |
| | CE-QUAL-W2 | 2001 | 4.6 µg/l | Brett et al., 2016 |
| | CE-QUAL-W2 | 2005-2010 | 6.7 to 13.2 µg/l | Kim et al., 2019 |
| | CE-QUAL-W2 | 2001-2011 | 62.9 µg/l | Brito et al., 2018 |
| | MINLAKE2020 | 2007-2009 | 0.6 - 27.6 µg/l | Tasnim et al., 2021 |
| | CE-QUAL-W2 | 2000-2019 | 19.36-25.57 µg/l | Almeida et al., 2023 |

## Code availability

The exact version of the models' source code is archived on Zenodo at https://doi.org/10.5281/zenodo.14606105 (Almeida and Coelho 2025). The current version of the open-source CEQUAL-W2 model (version 4.5) used in this study is also available from the project website (http://www.ce.pdx.edu/w2/, last access 24 January 2024).

## Data availability

Input files needed to run the models' and the hydrometric water quality and meteorological datasets used to force and validate each model are freely available and are archive on Zenodo at https://doi.org/10.5281/zenodo.14606105 (Almeida and Coelho 2025).

## Competing interests

The contact author has declared that none of the authors has any competing interests.

## Acknowledgements

The authors acknowledge the funding received from Fundação para a Ciência e a Tecnologia (FCT, Portugal), through the UIDB/04292/2020 and UIDP/04292/2020 strategic projects granted to the Marine and Environmental Sciences Centre 720 (MARE) and the LA/P/0069/2020 project granted to the Aquatic Research Network Associate Laboratory (ARNET).

**Author contribution:** MA conceptualized the study, developed the methodology, and handled software and data curation, as well as writing the original draft. PC administered the project and contributed to reviewing and editing the manuscript

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
