# Peer review of "Evaluating the performance of CE-QUAL-W2 version 4.5 sediment diagenesis model"

_Geoscientific Model Development, 2024_

## Author Response (AR1)

**RESPONSE TO REVIEWERS**

"Evaluating the performance of CE-QUAL-W2 version 4.5 sediment diagenesis model"
July 2025

Dear Editor-in-Chief

Thank you for the opportunity to submit a revised version of our manuscript, "Evaluating the Performance of the CE-QUAL-W2 Version 4.5 Sediment Diagenesis Model," for consideration in Geoscientific Model Development (GMD). We sincerely appreciate the time and effort you, the Associate Editor, and the Reviewers have devoted to evaluating our work. We are grateful for the insightful comments and constructive suggestions, which have significantly improved the quality of our manuscript.

We have carefully addressed all reviewer comments and incorporated the suggested revisions. Below, we provide a point-by-point response to each comment. For clarity, we refer to the revised manuscript without track changes, using page and line numbers (page–line) to indicate where modifications were made.

I look forward to hearing from you.

Sincerely,
Manuel Almeida

**Reviewer #1**

The authors have written an interesting, well-developed methods paper where they compare CE-QUAL-W2's zero-order sediment model against the full sediment diagenesis (SD) model introduced in V4 of the CE-QUAL-W2 water-quality model. I believe many water-quality modellers using CE-QUAL-W2 are reluctant to try the new SD model due to the sheer number of coefficients in the compartment, so it is interesting that the authors were able to model their waterbody mostly using the default parameters of the diagenesis model. While the authors primarily discussed the results for DO, there does seem to be value in collecting a few sediment samples where possible, based on the better results for TP, TN and (potentially) Chl-a with the SD model.

The introduction was good and provided sufficient context for why the authors thought the work was of interest to the water-quality modelling community.

**Author response:** We sincerely thank the reviewer for providing thoughtful feedback. We have carefully considered each of the comments and have made corresponding revisions to enhance the manuscript accordingly. We appreciate the recognition of our efforts to compare the zero-order sediment model with the full sediment diagenesis (SD) model in CE-QUAL-W2. We believe that for long-term studies, it is especially relevant to implement a more comprehensive sediment model. As the reviewer correctly noted, a key motivation for this study was to demonstrate that the SD model can produce reasonable and improved results even when using mostly default parameter values—an important consideration for practitioners who may be hesitant to adopt the model due to its complexity.

The methods were sufficient although the information regarding the configuration and calibration of the main water quality model could be more in-depth (e.g., appendix table of most the important coefficients) rather than leaving the reader to have to search through the CE-QUAL-W2 user manual.

**Author response:** Thank you for your comment. We agree that providing more detailed information on the configuration and calibration of the water quality model enhances the clarity and usefulness of the manuscript. In response, we have added tables A2 to A8 summarizing the most important coefficients and parameters used in the CE-QUAL-W2 model setup. This addition allows readers to understand the model calibration without needing to refer to the user manual. We believe this change improves the transparency and reproducibility of our modeling approach. The following sentence was included in the manuscript:

**PAGE 13 LINE 285**

"Tables A2 through A8 display the most significant CE-QUAL-W2 coefficients obtained after the calibration process."

I found one or two sections needed rereading several times to fully understand the objectives of the study and the model setup. Section 2.2 combines the model configuration (e.g., bathymetry, algal groups), a summary of the following method section, and a summary of the overall modelling approach, and I believe this could be better structured by separating the model set-up. Note that machine learning is not my

area of expertise and so I am unable to comment on the derivation of the forcing datasets for water-quality.

**Author response:** Thank you for your comment. We agree with the reviewer's suggestion. Accordingly, two new sections have been added: Section 2.2.1 – Model Setup and Section 2.3 – Modeling Approach. The Methods section has been revised as follows:

**PAGE 5 LINE 134-141**

2.2.1 Model Setup
The bathymetry of the Torrão reservoir was initially defined using a Digital Elevation Model (DEM) provided by Energies of Portugal, S.A. (EDP) and structured according to the methodology outlined in Wells (2021). The reservoir comprises one main branch (the Tâmega River), three tributaries and one distributed tributary (Fig. 1). Tributaries 1 and 2 are depicted in Fig 1. Tributary 3 represents the inflow from the Douro River into the pump-back system of the Torrão Reservoir. The bathymetric map includes 27 segments, each measuring 1000 meters in length, and a maximum number of 58 layers, each with a depth of 1 meter. Following this preliminary step, the reservoir boundary conditions (including water quality, hydrology, meteorology, and sediment characterization) were defined according to the methods described in Section 2.4. Due to the lack of available information, the model structure only includes a single algae group (Diatoms).

**PAGE 6 LINE 145 to PAGE 7 LINE 163**

2.3 Modeling approach
To thoroughly evaluate the capability of CE-QUAL-W2 in modeling dissolved oxygen using the sediment diagenesis module, the four available SOD modeling approaches were considered: Zero-order model; First-order model; Zero/First-order model (Hybrid model) and the sediment diagenesis model (SG model). The models were calibrated for the 2016–2021 period (see Section 2.5). During the results analysis, the performance metrics obtained during each model's calibration process were compared, along with the SOD values across the bottom layers of each model. A sensitivity analysis was conducted following calibration to evaluate each model's response: a) to varying POC, PON, and POP values in the case of the SG model; b) to different SOD values in the Zero-order and Hybrid models and c) to varying the initial first order sediment concentration in the case of the First-order model. Section 2.6 details the methodological approach used for the sensitivity analysis. To assess the sensitivity of each model to reductions in external organic matter (OM) and phosphorus ($PO_4$-P) inputs, two separate scenario analyses were conducted. The first scenario involved an 80% reduction in OM inflow load, while the second applied an 80% reduction in both OM and $PO_4$-P inflow loads. These reductions were implemented specifically in the main reservoir branch (Branch 1 – Tâmega River), where the majority of nutrient and organic inputs occur. Each sediment model—SD, Zero-order, First-order, and Hybrid—was run under baseline conditions and under both reduction scenarios. The impact on DO dynamics was evaluated using time series of depth- and segment-averaged DO concentrations. Each model—SD, Zero-order, First-order, and Hybrid—was run under baseline conditions and then under this reduced-loading scenario. The evaluation of model performance, along with the results

of the sensitivity analysis, provided deeper insights into simulating SOD dynamics using the sediment diagenesis approach in comparison to the other SOD formulations.

The results were well presented visually and with plenty of discussion provided by the authors. However, I was unable to follow what was being discussed and shown regarding TOC and POC in Section 3.3 (lines 310 to 325). It was not clear to me if the black line and circles show TOC or POC as the legend (TOC) and y-axis/caption (POC) are for different variables, nor could I follow how it was concluded that the particulate fraction of organic carbon constituted 40% of the TOC. Lines 310 to 320 and Figure 4 should be clarified.

**Author response:** Thank you for your comment. We agree with the reviewer that this section was not sufficiently clear. During the sediment characterization, we assumed that the POC was equal to the observed TOC value, primarily because POC was not directly measured. The simulation that used the POC value derived from the observed TOC data is Run 5, which was calibrated and referred to as the W2_SD model (Run 5 – baseline). Run 2 produced the best performance based on the NSE and RMSE criteria. The mean sediment concentration used to characterize Run 5 was 17,712 mg/L, calculated as the average of the following TOC values: 24,000; 20,064; 21,408; 19,296; and 5,376 mg/L (see Table 3). For Run 2, the mean value was 7,085 mg/L, based on the average of 9,600; 8,026; 8,563; 7,718; and 2,150 mg/L (also from Table 3). The value used in Run 2 (7,085 mg/L) is approximately 40% of the value used in Run 5 (17,712 mg/L). Therefore, if Run 5 assumes that POC is equal to the full TOC value, and Run 2 provides the best fit to observed data, it is reasonable to infer that the initial POC value should be approximately 40% of the observed TOC. Additionally, Figure 4 was corrected—the legend now reads initial POC value instead of initial TOC. This section has been revised as follows to improve clarity. Please note that an issue was identified with the initial metric estimates, and all performance metrics were recalculated accordingly. The manuscript results have been updated to reflect this correction.

**PAGE 20 LINE 351-371**

"The SOD values strongly influence the water column DO; therefore, this parameter was considered to support this analysis. Figure 7 shows the SOD values from the reservoir bottom layer, predicted by the SD model for Runs 1 to 6, compared with the RMSE (Fig7A) and the NSE (Fig7B) values obtained between the predicted water column DO profiles and the mean initial POC values (across all sites values) for each run. These results suggest that Run 4 was the best modeling solution. Considering the results obtained for Run 5 (baseline), Run 4 reduced the RMSE from 2.015 mg/L (Run 5) to 2.011 mg/L (Run 4) and increased the NSE from 0.714 (Run 5) to 0.716 (Run 4). The average SOD value in the bottom layer of the reservoir (across all model segments) decreased from 1.162 g $O_2$/m²day (Run 5) to 1.071 g $O_2$/m²day (Run 4). Although the reduction is modest and had only a minor effect on the DO profile predictions (Fig. 9), it suggests that the initial POC values used in Run 5 were likely overestimated. This outcome aligns with the assumption made in Run 5, where all observed TOC was considered to exist entirely as POC. In contrast, Run 4 was characterized using a lower average sediment concentration. Specifically, the mean

value used in Run 4 (14170 mg/L) represents approximately 80% of the TOC value used in Run 5 (17712 mg/L), which was derived from observed TOC measurements (see Table 3). This comparison suggests that a more realistic estimate is that about 80% of the total organic carbon exists in particulate form, with the remainder composed of dissolved organic carbon. Run 4 and Run 5 show negligible differences in the predicted water temperature and DO profiles (Fig. 8 and 9). Table A10 presents the performance metrics for water temperature, DO, TN, TP, $BOD_5$, and Chl-a obtained for Run 4. While this run improved the DO simulation in the reservoir, results for the other constituents remained very similar to those of Run 5 (baseline). Overall, the water temperature profiles are very well captured by all models (Fig. 8), reflecting their robustness in simulating thermal dynamics. In contrast, DO profiles are more complex and challenging to model due to their sensitivity to multiple interacting processes. Nevertheless, the models were able to capture the main seasonal and vertical trends in DO concentrations, including stratification patterns and general oxygen depletion in bottom layers during warmer months (Fig.9)."

Furthermore, while this paper is of interest for those of us using the CE-QUAL-W2 model, and could be cross-transferred to other waterbodies using the CE-QUAL-W2 model, the authors did not attempt to place their findings in the context of the broader water-quality modelling science, and how this work may contribute. I think this should be added to the discussion to strengthen this submission.

**Author response:** Thank you for this comment. We appreciate your suggestion to broaden the context of our findings within the field of water-quality modeling science. In response, we have revised the discussion to clarify how our study contributes more broadly to sediment oxygen demand modeling in CE-QUAL-W2 and to the wider field of water-quality modeling.

We now emphasize that while the study's primary focus was to evaluate the performance of the sediment diagenesis (SD) model, the inclusion of alternative formulations (Zero-order, First-order, and Hybrid models) not only allowed for a direct performance comparison but also provided practical insights into model applicability under varying system conditions. We discuss the relative strengths and limitations of each approach, emphasizing how their performance relates to model structure, data availability, and application scale (e.g., short- vs long-term simulations).

Additionally, we highlight how the findings align with broader principles in ecological and environmental modeling, such as model parsimony (Burnham and Henderson, 2002) and user expertise (Piccolroaz et al., 2024). These insights are transferable to other water bodies and modeling frameworks, particularly where users face similar trade-offs between model complexity and data constraints. These revisions aim to better position the study within the broader water-quality modeling literature and demonstrate its relevance beyond the specific application to our study reservoir.

**PAGE 31 LINE 538-561**
It is important to emphasize that this study was primarily designed to evaluate the performance of the sediment diagenesis model. However, by incorporating alternative SOD modeling approaches, it inevitably allowed for a comparative ranking of model performance, highlighting the relative strengths and limitations of each formulation. The performance limitations of the Zero-order and First-order models can be attributed to their structural simplifications. Specifically, the Zero-order model's strong

temperature dependence, coupled with its disregard for the dynamics of organic matter loading, reduces its ability to capture temporal variability driven by external inputs. Similarly, the lower accuracy of the First-order model likely stems from its exclusion of anaerobic decay processes and limited representation of sediment biogeochemistry, which becomes especially relevant under low-oxygen conditions. The Hybrid model outperformed all other approaches. Considering the principle of parsimony (Occam's razor) (Burnham and Henderson, 2002), the simpler Hybrid model proved more effective than the complex SD model, making it the preferred choice for simulating SOD dynamics in the reservoir. These findings underscore the importance of selecting models that align with the specific characteristics of the system being studied. Simpler models, such as the Hybrid model, may be adequate for steady-state conditions, short- to medium-term forecasts, or scenarios with limited data. The zero-order SOD component of the Hybrid model relies solely on temperature and is decoupled from the water column; therefore, in long-term simulations, this limitation can gradually undermine the model's accuracy. In contrast, the SD model may be more appropriate when the goal is to explore system-wide feedbacks and temporal dynamics over extended periods—especially those involving sediment accumulation and nutrient cycling—where it may provide valuable insight into underlying processes, provided that sufficient observational data become available to support its additional state variables. Moreover, a model's effectiveness heavily depends on the user's familiarity with its structure and their skill in calibration. Yet, it is unrealistic to expect researchers to master the implementation of every available modeling approach. As such, comparisons between models should be interpreted carefully, acknowledging the influence of user expertise on performance outcomes (Piccolroaz et al. 2024). Overall, to strengthen the analysis, it is recommended that users apply all available SOD modeling approaches in the case of the CE-QUAL-W2 model and assess the model's behavior. This comprehensive evaluation provides a solid foundation for further modeling efforts and helps ensure that the chosen approach is well-suited to the system's specific conditions and objectives.

Finally, there were numerous editorial errors throughout the manuscript that need addressing; a few examples below, although there are more:
1) Discrepancies in the citations and the bibliography. Examples include:
Line 54: Should be just 'Zoubabi-Aloui'
Line 73: I believe this should be 'Wells 2021'
Line 139: 'Adelena et al. 2015', does not appear in the bibliography
Line: 142: Should be 'Berger and Wells 2014'
Etc.
**Author response:** Thank you for bringing this to our attention. We have carefully reviewed the entire manuscript and addressed the editorial issues you noted, including correcting the discrepancies between in-text citations and the bibliography. Specifically:

Line 65: Corrected to 'Zouabi-Aloui'

Line 73: The sentence with this reference was removed.

Line 139: Removed 'Adelena et al. 2015' as it does not appear in the bibliography

Line 126: Corrected to 'Berger and Wells 2014'

In addition, we have conducted a thorough review to identify and fix any remaining citation and formatting inconsistencies throughout the manuscript and reference list. We appreciate your careful reading and helpful comments.

2) Also seems to be some discrepancies in the Section number cross-refs (for example Lines 106 and 109, refer to Section 1.2.3 and 1.2.4, respectively, with other instances throughout the document).
**Author response:** Thank you for pointing this out. The section numbers have been corrected accordingly.

3) Line 285 .. for DO, "…the W2_zero-order model performed slightly better according to all metrics, with the exception of PBIAS". I am wondering if the authors mean R2 (which is marginally worse than the SD model)? Perhaps it is me that is mistaken, but for PBIAS it seems the zero-order model performs better for DO than the SD model, with the assumption the goal is a low-bias model. This should be clarified.
**Author response:** Thank you for pointing this out. You are correct to note the inconsistency. Following the inclusion of three additional SOD models and a recalculation of the performance metrics, we have revised the sentence in question to reflect the updated results more accurately. The original statement has been replaced with the following text to clarify the comparative performance of the models with respect to DO, including a corrected interpretation of PBIAS and $R^2$ values.

**PAGE 13 LINE 286 to PAGE 14 LINE 312**

"Tables A2 through A8 display the most significant CE-QUAL-W2 coefficients obtained after the calibration process. The results of the calibration process for all models, are presented in Table 4 and Table A9 and illustrated in figures 3 to 6 and figures 8 and 9. The performance metrics for water temperature across the different sediment models show consistent accuracy, with NSE and $R^2$ values ranging from 0.95 to 0.96 and minimal variation across models. The RMSE and MAE for temperature also remain low, indicating reliable thermal performance regardless of the sediment model applied. In contrast, DO predictions show more variability. The Hybrid model achieved the best overall DO performance, with the highest NSE (0.76 ± 0.30) and $R^2$ (0.76 ± 0.31), as well as the lowest RMSE (1.87 ± 0.72) and MAE (1.22 ± 0.55), while maintaining a near-zero PBIAS (-0.55 ± 11.14), indicating minimal systemic bias. The Zero-order model also performed reasonably well, with slightly lower error metrics than the SD model. The First-order model, however, showed the weakest DO performance, with a lower NSE (0.68 ± 0.22), higher RMSE (2.15 ± 0.82), and a significant negative PBIAS (-12.17 ± 15.44), suggesting an underestimation of oxygen concentrations. Overall, the results suggest that while temperature simulation is robust across all models, DO dynamics are better captured using the Hybrid or Zero-order models, with the Hybrid model offering the most balanced and accurate representation under the tested conditions. However, the differences in performance metrics for DO among the models are relatively small and often fall within overlapping standard deviations, with the exception of the First-order model, which consistently shows lower accuracy and higher bias, suggesting that while the Hybrid model offers slightly better overall performance, the improvements over the SD and Zero-order models are modest and should be interpreted with caution. In terms of nutrient dynamics, the Hybrid and Zero-order models improve TN and TP predictions relative to the SD and First-order models.

The Hybrid model, for example, improves TN R² to 0.31 and TP to 0.27, although the associated biases remain significant (e.g., −18.75% for TN and +36.49% for TP). $BOD_5$ and Chl-a remain poorly simulated across all models, with R² values consistently low (≤0.06 for Chl-a and ≤0.03 for $BOD_5$), and large PBIAS values, particularly in the SD and First-order configurations. The Zero-order model slightly reduces bias in Chl-a and Total N compared to the SD model but performs poorly for TP due to a large overestimation (PBIAS = 103.43%) (Fig.4D). Notably, the SD and First-order models failed to reproduce observed phosphorus release events from sediments on 2018-09-18, 2020-09-08, and 2021-08-31 (Figures 3D and 5D). In contrast, the Hybrid model successfully captured these events by modeling phosphorus release as a linear function of SOD, providing a more realistic representation of sediment–water nutrient interactions (Fig.6D). Overall, while no model fully captures the complexity of all constituents, the Hybrid model consistently provides the most balanced and improved representation, particularly for DO and nutrient parameters."

4) Line 312: It should read Fig4b after NSE. **Author response:** Thank you for point this out. The sentence was corrected.

**PAGE 20 LINE 351-354**

"The SOD values strongly influence the water column DO; therefore, this parameter was considered to support this analysis. Figure 7 shows the SOD values from the reservoir bottom layer, predicted by the SD model for Runs 1 to 6, compared with the RMSE (Fig7A) and the NSE (Fig7B) values obtained between the predicted water column DO profiles and the mean initial POC values (across all sites values) for each run."

**Reviewer #2**

Comments on Evaluating the performance of CE-QUAL-W2 version 4.5 sediment diagenesis model Manuel Almeida, Pedro Coelho
Overall, this is a useful evaluation of the sediment diagenesis model in CE-QUAL-W2 model. The next logical step would be to compare first order and zero order model with sediment diagenesis. The MAE for temperature simulations seems high compared to other systems and this can drastically affect dissolved oxygen profiles. This may be the result of inflow temperatures as well as outflow dynamics. It would be useful to work on improving temperature predictions (if there is a path forward) and to see how that affects the results in this study. The dissolved oxygen profiles are very complex in this reservoir and often the model reproduced the correct shape of the profiles.

**Author response:** We appreciate the time and effort that the reviewer has invested in evaluating our manuscript. Their insightful comments and constructive suggestions have been invaluable in helping us improve the quality and clarity of our work. We have addressed the reviewer's suggestion, and the revised manuscript now includes four distinct modeling approaches: (i) a user-defined zero-order model, (ii) a simple predictive first-order model, (iii) a hybrid approach combining the zero- and first-order models, and (iv) the sediment diagenesis model. While revising the manuscript, we discovered that the calibration metrics had not been properly applied. After correcting this, the mean absolute error (MAE) for water temperature across all simulations is now $0.88\,°C \pm 0.02\,°C$, which can be considered a very reasonable value. A new figure—Figure 8—was included to show the observed and predicted water temperature profiles, allowing for a clearer comparison of model performance across depths and time. We agree with the reviewer that the dissolved oxygen profiles in the reservoir are quite complex. However, the discrepancies between the modeled and observed dissolved oxygen concentrations are primarily driven by factors other than water temperature—namely, the inflow of organic matter and algal biomass. Since the boundary conditions are the same across all models and the models reproduce the dissolved oxygen profiles reasonably well, we believe that the modeling approach is both sound and well-substantiated.

[Figure]

Figure 8: Observed water temperature profiles (300 m from the dam) compared to predicted profiles using the SD model (Run 4) and (Run 5 - baseline), Zero-order model (zero-order SOD = 2.5 g O2/m2day - baseline); First-order model (ISC= 0.5 g/m² - baseline) and the Hybrid model (zero order SOD= 1.0 g O2/m2day - baseline).

There were a few comments on the text which are summarized below:
Line 42-43: "if the SOD is not accurately computed the waterbody phosphorous balance will, in turn, be incorrect." This expression needs further explanation. If the zero order SOD model is used, then the anoxic release of PO4 is a linear function of the SOD in the CE-QUAL-W2 model, in other words SOD[g O2/m2/day]*PO4release rate [g P/g O2]. If one uses a predictive model, like sediment diagenesis, then the SOD and P release from the sediments will be a function of the organic and nutrient loading of particulate matter from the water column.

**Author response:** Thank you for pointing this out. We agree with the reviewer that this section was unclear and have revised the text accordingly, as follows:

**PAGE 2 LINE 42-60**

"The main challenge with these modeling approaches is that the sources of DO depletion—such as the inflow of organic matter or algal mortality—can significantly influence DO dynamics, and these sources must be well characterized to ensure accurate predictions. While the baseline model can reproduce observed DO profiles with reasonable accuracy, its predictive reliability may be compromised if key DO sinks and sources are not well defined.

For example, the model's response to a reduction in external phosphorus loading is influenced by internal phosphorus release from sediments during anoxic periods. In CE-QUAL-W2, when a zero-order SOD model is used, the anoxic release of phosphate ($PO_4$) is modeled as a linear function of SOD: SOD [g $O_2$/m²day] × $PO_4$ release rate [g P/g $O_2$]. Thus, any error in the estimation of SOD will directly affect the predicted internal phosphorus loading, and by extension, the overall phosphorus balance in the waterbody. In contrast, when using the predictive sediment diagenesis model, internal phosphorus loading depends on the organic and nutrient inputs from particulate matter in the water column and the sediment's biogeochemical response, which is highly influenced by the initial value of particulate organic carbon (POC). As a result, this approach introduces additional uncertainty when key particulate components are not adequately measured or constrained in both the water column and sediments. Calibrating other constituents, such as orthophosphate (P-$PO_4$), can help reduce uncertainty. P-$PO_4$ is released from sediments under anaerobic conditions, and its calibration can enhance the accuracy of DO modeling. Still, this release is influenced by multiple factors, including the initial sediment P-$PO_4$ concentration and the release rate (in the zero-order model), or the mineralization of POP (in the diagenesis model). In both cases, significant uncertainty remains without observed data for POC, PON, and POP in both the water column and sediments. Of these, POC has the most significant influence on SOD, making access to sediment POC data essential for improving model accuracy, even when PON and POP measurements are lacking."

Line 48: "In other words, the modeling uncertainty may diminish but will persist without observed POC, PON and POP" – it is unclear, is this a discussion about water column POC, PON and POP or sediment POC, PON, and POP?
**Author response:** We appreciate the reviewer's comment and acknowledge that the original sentence lacked clarity. We believe that the previous revised sentence (PAGE 2 LINE 43-61), addresses this concern by clearly referring to the need for observed POC, PON, and POP data in both the water column and sediments.

Line 73: "dissolved oxygen uptake rates in the water column (Wells, 2011)." – reference to Wells, 2011 not found in references~
**Author response:** Thank you for pointing this out. The reference was corrected to Wells, 2021.

Line 114-115: "This is not, however, a predictive approach, as, other than variations resulting from the temperature dependence of the decay rate, the rates remain constant over time (Wells, 2021)." – Note that also when there is anoxia in the water column SOD is turned OFF.

**Author response:** Thak you for you comment. The following sentence was included in the manuscript.

**PAGE 4 LINE 111-113**

"The zero-order model is not a predictive approach, as, other than variations resulting from the temperature dependence of the decay rate, the rates remain constant over time (Wells, 2021). Additionally, under anoxic conditions in the water column, SOD is disabled in the model."

Line 142: "model has been elaborated in works by Prakash et al. (2014), Berg and Wells (2014), and Vandenberg et al. (2015)" – change 'Berg' to 'Berger'. Also, the V4.5 model had many enhancements to the sediment diagenesis model as outlined in the User Manual. The initial V4 model is much different and limited compared to the V4.5 model. **Author response:** Thank you for pointing this out. The following sentence was included in the manuscript:

**PAGE 4 LINE 123 to PAGE 5 LINE 126**
"The conceptual framework of the model has been elaborated in works by Prakash et al. (2014), Berger and Wells (2014), and Vandenberg et al. (2015). It is important to note that significant enhancements to the sediment diagenesis module were introduced in version 4.5 of the model, as detailed in the User Manual (Wells, 2021)."

Line 157-158: "The meteorological data used to drive the model, including hourly air temperature, dew point, solar radiation, cloud cover, and wind characteristics, were sourced from ERA5-Land…" – Was there an effort to ground-truth this ERA5-Land dataset with on-site meteorological measurements in the area as a check?
**Author response:** Thank you for pointing this out. Unfortunately, there are no on-site meteorological stations within the study region available to directly validate the ERA5-Land dataset. However, at the initial stage of the study, we referred to the findings of Almeida and Coelho (2023b), "A First Assessment of ERA5 and ERA5-Land Reanalysis Air Temperature in Portugal," and Barbosa et al. (2022), "Extreme Heat Events in the Iberian Peninsula from Extreme Value Mixture Modeling of ERA5-Land Air Temperature." Their analyses demonstrated a strong correlation between observed and reanalysis air temperature data at both daily and seasonal timescales, supporting the reliability of ERA5-Land data in this region. Moreover, the model's performance metrics for water temperature prediction further support the adequacy of the meteorological forcing, indicating that it was appropriately captured and contributed to the accurate simulation results. The following sentence was added to the manuscript to reflect this clarification:

**PAGE 7 LINE 165-171**
"The meteorological data used to drive the model, including hourly air temperature, dew point, solar radiation, cloud cover, and wind characteristics, were sourced from ERA5-Land, a high-resolution reanalysis dataset optimized for land applications.

Although no on-site meteorological stations are available in the study area for direct validation, studies by Almeida and Coelho (2023b) and Barbosa et al. (2022) have demonstrated a strong correlation between ERA5-Land air temperature data and observed measurements at regional scales, supporting the reliability of this dataset for our modeling purposes. Furthermore, the accuracy of water temperature predictions in our simulations indicates that the meteorological forcing was well represented, confirming the suitability of ERA5-Land data for driving the model."

Line 235: "six state variables was evaluated with five different metrics (vide section 1.2.6)." – note sure what '(vide section' means – typo?
**Author response:** Thank you for pointing this out. We intended "vide" to direct the reader to Section 1.2.6 for further details. However, we recognize that this usage may be unclear or unfamiliar to some readers. To improve clarity, we have replaced "vide" with "see" in the revised manuscript.

Line 245: "two parameters retained their default values shown in Table 1." – I think Table 1 is an incorrect table reference.
**Author response:** Thank you for pointing this out. You are correct—the correct table reference is Table 3, not Table 1. We have updated the manuscript accordingly.

Line 305: Figure 3 is very hard to see data and model. Figure needs to be broken up or redone to allow others to see model vs data clearly.
**Author response:** Thank you for pointing this out. We have included one figure per model.

[Figure]

Figure 3: Constituents observed values at three different depths: (a) an integrated sample between the reservoir surface and an average depth of 5.8 meters, (b) an average depth of 23 meters, and (c) an average depth of 43.7 meters. These observed values were compared with the predicted time series from the SD model (run 5 - baseline) (A to F) for the same depths.

[Figure]

Figure 4: Constituents observed values at three different depths: (a) an integrated sample between the reservoir surface and an average depth of 5.8 meters, (b) an average depth of 23 meters, and (c) an average depth of 43.7 meters. These observed values were compared with the predicted time series from the Zero-order model (zero order SOD = 2.5 g O2/m2day - baseline) (A to F) for the same depths.

[Figure]

Figure 5: Constituents observed values at three different depths: (a) an integrated sample between the reservoir surface and an average depth of 5.8 meters, (b) an average depth of 23 meters, and (c) an average depth of 43.7 meters. These observed values were compared with the predicted time series from the First-order model (ISC=0.5 g/m2 - baseline) (A to F) for the same depths.

**Hybrid model (zero-order SOD = 1.0 g O$_2$/m$^2$/day - baseline)**

[Figure]

Figure 6: Constituents observed values at three different depths: (a) an integrated sample between the reservoir surface and an average depth of 5.8 meters, (b) an average depth of 23 meters, and (c) an average depth of 43.7 meters. These observed values were compared with the predicted time series from the Hybrid model (zero order SOD= 1.0 g O2/m2day - baseline) (A to F) for the same depths

Line 391: "W2_zero-order model (2.50 $gO_2/m^2/day$) was significantly higher than the mean SOD computed with the best W2_SD model (Run 2) (0.810 $gO_2/m^2/day$)." – This is not a correct comparison since the Zero order SOD was at 20oC (or at its maximum) and the SD model result is actual SOD at the temperature at the bottom of each segment. Looking at the temperature near the bottom in Fig 3 a year-round average is probably around 10-12oC – hence much lower year-round than the 20oC maximum rate.

**Author response:** Thank you for pointing this out. The reviewer is correct—we inadvertently used fixed maximum rates for zero-order SOD instead of representing it as a function of temperature. In the revised version we compare the temperature-corrected zero-order SOD value (using bottom water temperature) with the SOD flux from the sediment diagenesis model as applied to the bottom water layer, since this reflects the total sediment oxygen demand at the sediment-water interface.

Line 392: "This can be explained by the fact that the W2_zero-order model SOD represents all of the reservoir's DO uptake rate in the water column and not just the sediment uptake." – See comment above – it is related to the temperature. The zero order model only is for sediment demand, not water column demand.

**Author response:** Thank you for pointing this out. The reviewer is correct—this sentence, as written, is inaccurate. We acknowledge that the zero-order SOD model specifically represents oxygen consumption at the sediment-water interface, and does not account for other oxygen-demanding processes such as BOD decay or nitrification in the water column, which are modeled separately. Our original intention was to suggest that, conceptually, a higher SOD value might reflect the overall oxygen demand, including contributions from other sources influencing oxygen uptake. However, as previously mentioned, the zero-order SOD was initially not computed using bottom-layer temperature, making the interpretation misleading. Therefore, this sentence is no longer valid and has been removed from the revised manuscript.

Line 400: "The zero-order model employs a constant SOD value that only varies with water temperature and does not account for organic matter decay or its impact on SOD values." – Why did you not use the zero order model with the first order model as reported in your introduction?

**Author response:** Thank you for the comment. In this study, we compared the performance of the zero-order sediment model and the sediment diagenesis model in simulating observed dissolved oxygen profiles. While CE-QUAL-W2 allows for the simultaneous use of zero-order and first-order sediment compartments, we initially chose not to include the first-order model, as it was not essential to our original research objective. Our goal was to evaluate the performance of two contrasting modeling approaches: the zero-order model, which is the simplest representation of sediment oxygen demand in CE-QUAL-W2, and the sediment diagenesis model, which is the most detailed. This choice allowed us to assess model behavior across the spectrum of complexity—from a highly simplified empirical approach to a more process-based, predictive framework. The first-order model, although it provides a dynamic response to increased organic matter flux to the sediments, does not simulate nutrient release processes such as phosphorus release. Representing such processes would require coupling it with the zero-order model, introducing additional

complexity and interaction effects that were beyond the scope of our initial comparison. However, following the reviewer's suggestion, we have revised the manuscript to include two additional modeling approaches involving the first-order model. The revised manuscript now evaluates four distinct sediment modeling configurations: (i) a user-defined zero-order formulation decoupled from the water column, (ii) a simple predictive first-order model, (iii) a hybrid approach combining zero- and first-order models, and (iv) the sediment diagenesis model. The manuscript has been updated accordingly to reflect these additions and all models were made available in Almeida, M., and Coelho, P., 2025. Furthermore, a new section—3.4 Inflow Organic Matter and Phosphorus Load Reduction Scenarios—was added to assess the sensitivity of each model to reductions in external inputs of organic matter (OM) and phosphorus ($PO_4$-P). Two separate scenario analyses were conducted: the first involved an 80% reduction in OM inflow, and the second applied an 80% reduction in both OM and $PO_4$-P inflow loads. These reductions were implemented specifically in the main reservoir branch (Branch 1 – Tâmega River), which receives the highest nutrient and organic inputs. Accordingly, the Methods section was updated to reflect these new scenarios.

**PAGE 1 LINE 7-23**

[revised manuscript text omitted]